# MEDIQ: Question-Asking LLMs and a Benchmark for Reliable Interactive Clinical Reasoning

**Shuyue Stella Li**[1]    **Vidhisha Balachandran**[2]    **Shangbin Feng**[1]    **Jonathan S. Ilgen**[1]
**Emma Pierson**[3]    **Pang Wei Koh**[1,4]    **Yulia Tsvetkov**[1]
[1]University of Washington    [2]Carnegie Mellon University    [3]Cornell Tech
[4]Allen Institute for AI
stelli@cs.washington.edu
https://github.com/stellalisy/mediQ
https://huggingface.co/datasets/stellalisy/mediQ

## Abstract

Users typically engage with LLMs interactively, yet most existing benchmarks evaluate them in a static, single-turn format, posing reliability concerns in interactive scenarios. We identify a key obstacle towards reliability: LLMs are trained to answer any question, even with incomplete context or insufficient knowledge. In this paper, we propose to change the static paradigm to an interactive one, develop systems that *proactively ask questions* to gather more information and respond reliably, and introduce an benchmark—**MEDIQ**—to evaluate question-asking ability in LLMs. MEDIQ simulates clinical interactions consisting of a Patient System and an adaptive Expert System; with potentially incomplete initial information, the Expert refrains from making diagnostic decisions when unconfident, and instead elicits missing details via follow-up questions. We provide a pipeline to convert single-turn medical benchmarks into an interactive format. Our results show that directly prompting state-of-the-art LLMs to ask questions *degrades* performance, indicating that adapting LLMs to proactive information-seeking settings is nontrivial. We experiment with abstention strategies to better estimate model confidence and decide when to ask questions, improving diagnostic accuracy by 22.3%; however, performance still lags compared to an (unrealistic in practice) upper bound with complete information upfront. Further analyses show improved interactive performance with filtering irrelevant contexts and reformatting conversations. Overall, we introduce a novel problem towards LLM reliability, an interactive MEDIQ benchmark and a novel question-asking system, and highlight directions to extend LLMs' information-seeking abilities in critical domains.

## 1   Introduction

General-purpose large language models (LLMs) are designed to serve a broad audience by following instructions and providing the most likely and general answers (Brown et al., 2020; Ouyang et al., 2022). However, in high-stakes decision making scenarios such as clinical conversations, LLM assistants can be harmful if they provide general responses instead of gathering missing information to make informed decisions. As shown in Figure 1, standard medical question-answering (QA) tasks are formulated in a single-turn setup where all necessary information is provided upfront, and the model is not expected to interact with users (Jin et al., 2021; Pal et al., 2022; Jin et al., 2019; Hendrycks et al., 2020). This QA paradigm diverges from real-world scenarios, where users may provide **incomplete information**, and effective decision-making often requires an **investigative process** involving follow-up questions to clarify and gather necessary details (Trimble & Hamilton, 2016; Bornstein & Emler, 2001; Masic, 2022).

38th Conference on Neural Information Processing Systems (NeurIPS 2024).

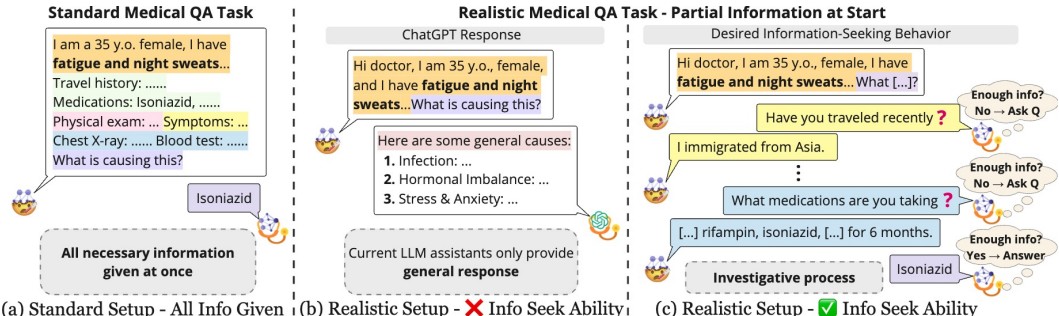

**Figure 1:** Information Seeking Task. In standard medical QA tasks (left), all necessary information is given to the assistant model at the same time. When given partial information, current LLMs only provides general responses (middle). In a more realistic scenario (right), the presentation of patient information relies on proactive elicitation from the doctor; our proposed MEDIQ framework operationalizes this scenario.

This gap between existing benchmarks and reality calls for a paradigm shift to designing systems adept at navigating high-stakes interactive scenarios. Focusing on clinical interactions where context is often incomplete, we introduce **MEDIQ**, an interactive benchmark for **m**edical **e**valuation with **d**ynamic **i**nformation-seeking **q**uestions, to address limitations of static single-turn QA benchmarks (Figure 2). Unlike conventional systems, which assume that all necessary information is readily available, MEDIQ acknowledges the inherent uncertainty in medical consultations where a typical patient does not have the expertise to distill all necessary and relevant information they need to provide. To achieve this, MEDIQ comprises two components: a **Patient system** that simulates a patient and responds to follow-up questions, and an **Expert system** that serves as a doctor's assistant and asks questions to the patient before making a medical decision. In this **interactive clinical reasoning task**, a successful information-seeking Expert should decide, at each turn, whether it has enough information to provide a confident answer; if not, it should ask a follow-up question.

We convert two medical QA datasets, MEDQA (Jin et al., 2021) and CRAFT-MD (Johri et al., 2023, 2024), into an interactive benchmark by parsing the patient records to only provide partial information in the beginning. We first develop and validate a Patient system that accurately answers Expert inquiries by retrieving the correct facts from the patient record. We then benchmark Expert systems based on state-of-the-art (SOTA) LLMs, including Llama-3 (Touvron et al., 2023), GPT-3.5 (Brown et al., 2020) and GPT-4 (OpenAI et al., 2024), to evaluate their proactive information seeking ability. It is striking that prompting these models to ask questions results in an 11.3% accuracy drop compared to starting with the same limited information and asking no questions, showing that adapting LLMs to interactive information-seeking settings is nontrivial. A key challenge is deciding when to ask a follow-up question instead of directly providing an answer. With confidence estimation strategies such as rationale generation and self-consistency, we improve Expert performance by 22.3%, although a 10.3% gap remains compared to an upper bound when full information is presented at once.

Our results show that while SOTA LLMs perform relatively well with complete information, they struggle to proactively seek missing information in a more realistic, interactive settings with incomplete initial information. By providing a modular, interactive benchmark, we hope to facilitate the development of reliable LLM assistants for complex decision-making in healthcare and other high-stakes domains. Our main contributions are:

1. We identify the critical problem of **information-seeking questions** in reliable interactive LLM assistants. We propose a paradigm shift and a practical conversion pipeline from standard single-turn benchmarks into interactive settings with incomplete initial information.

2. We develop the **MEDIQ Benchmark** to simulate more realistic clinical interactions between a Patient System and an Expert System. We rigorously develop and test the Patient System to benchmark any Expert's information-seeking and clinical decision-making abilities.

3. We show that SOTA LLMs such as Llama-3-Instruct, GPT-3.5 and GPT-4 struggle at proactive information seeking, revealing a significant gap in this area.

4. We propose **MEDIQ-Expert**, our best Expert system with novel abstaining capabilities to reduce unconfident answers, to partially close the gap between the more realistic incomplete information setup and the existing full information setup.

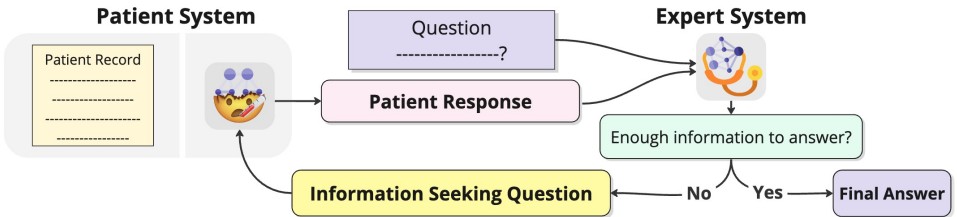

**Figure 2:** The MEDIQ Benchmark. MEDIQ operationalizes a more realistic dynamic clinical interaction between a Patient system and an Expert system to evaluate info-seeking and question-asking.

## 2  MEDIQ: Dynamic Medical Consultation Framework Overview

**Task Definition**    The dynamic medical consultation task simulates the iterative nature of real-world clinical interactions. This task starts by providing an initial patient description $k_0$ of their conditions to the Expert system. The initial information typically contains the patient's age, gender, and chief complaint for the visit. The Patient system has access to the entire patient record $\mathcal{K} = \{k_0, k_1, \ldots, k_n\}$, and the necessary information to answer the multiple choice question is $\mathcal{K}^* \subseteq \mathcal{K}$. At the start of the $t$-th turn, the knowledge available to the Expert system is denoted as $\mathcal{K}_{t-1} = \{k_0, \ldots, k_i\}$. Given follow-up question $q_t$, the Patient system responds with $r_t = \{k | k \in \mathcal{K}\}$. The Expert knowledge is then updated as $\mathcal{K}_t = \mathcal{K}_{t-1} \cup r_t$. The **main challenge of the task** is for the Expert system to ask information-seeking questions to expand $\mathcal{K}_t$ until the knowledge gap is filled, i.e. $\mathcal{K}_t = \mathcal{K}^*$, at which point the Expert system is asked to make a final decision.

### 2.1  The Patient System

**Patient Task**    As part of the MEDIQ framework, the Patient system simulates a human patient in clinical conversations. The Patient system has access to the full patient record that is sufficient for the diagnosis, including symptoms, onset duration, medical history, family history, and/or relevant lifestyle factors. The Patient system uses the patient record and a single information-seeking question from the Expert system to produce a coherent response consistent with the given patient information as shown in Figure 2. A reliable Patient system is critical to simulate a real and accurate medical consultation process. We propose that any Patient system should be evaluated on (1) *Factuality* - measuring if a patient's responses are faithful to the patient's record and history and (2) *Relevance* - measuring if the patient's response answers the expert's question. Given the full patient record and the expert question, we propose and evaluate three Patient system variants: **Direct**, **Instruct**, and **Fact-Select**, to obtain the patient response. Exact prompts and examples are in Appendix A.2.

1. **Direct:** Serving as a baseline, the Patient treats the response-generation as a reading comprehension task with no additional instruction. The prompt includes the patient's record followed by the Expert's question and asks the model to directly respond to the question using the given paragraph.
2. **Instruct:** The Patient is instructed to respond truthfully to the Expert's question using the patient record only. When the context does not contain an answer to the question, the Patient is instructed to refrain from answering.
3. **Fact-Select:** The Patient aims to improve the factuality of the response by decomposing the patient record into atomic facts and responds by selecting facts that are relevant to the Expert's question.

### 2.2  The Expert System

**Expert Task**    The Expert system simulates the medical decision-making process of experienced clinicians, who seek additional patient information and iteratively update their differential diagnosis. The Expert system is first presented with a medical question and limited patient information. As each turn, it assesses whether the provided information is sufficient to answer the question. If the Expert system is unconfident, it can elicit evidence with a follow-up information-seeking question; otherwise, the Expert system deems the acquired information sufficient and provides a final answer. The performance of the Expert system is evaluated on the (1) *efficiency* of the conversation (number of follow-up questions) and (2) the *accuracy* of the final diagnosis.

#### 2.2.1  Expert System Breakdown

Medical decision making is a complex process involving clinical reasoning and proactive information-seeking (Bordage, 1999; Norman, 2005; Schmidt et al., 1990; Boshuizen & Schmidt, 1992; Patel

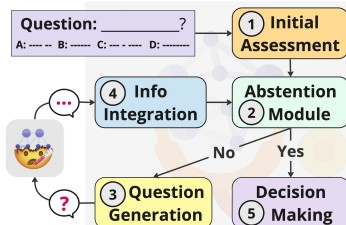

**Figure 3:** Expert system information flow breakdown.

et al., 1994). We describe our proposed **MEDIQ-Expert**, which operationalizes the Expert system by breaking down the task into five medically-grounded steps: (1) initial assessment, (2) abstention, (3) question generation, (4) information integration, and (5) decision making (Figure 3). Each step is modular and easily modifiable.

**Step 1. Initial Assessment Module**: Given limited patient intake information and the multiple choice question (MCQ) as the input, the goal of this module is to provide an initial assessment of the patient. The Expert system is asked to produce a paragraph that elaborates on the symptoms and options, and identifying potential knowledge gaps (e.g., additional symptoms, lab tests) missing for answering the question. This step is done only once at the beginning of the interaction, and we keep the output in the conversation thread for future turns to refer back to.

**Step 2. Abstention Module**: When the model is not confident, it should *abstain* from giving an answer and asks a information-seeking question instead. The goal of the Abstention Module is to evaluate the **confidence level** of the Expert system to make a decision given the available information. The input to this module is the MCQ and the patient information consisting of the initial presentation and a conversation log of follow-up questions and responses. We probe the confidence level of the model to reliably answer the question via prompting (§ 2.2.2). The output of this module is a yes/no answer for whether to proceed to final answer. If the model is confident, it skips to decision making; otherwise, it continues to question generation.

**Step 3. Question Generation Module**: When more information is deemed necessary, the goal of the question generation module is to craft an information-seeking question to elicit additional medical evidence such as lifestyle factors and physical exam results. The input to this module is all previous reasoning steps, and the acquired patient information; the notion of atomic questions is defined with respect to the medical domain in the prompt, and the output is an atomic question to the patient.

**Step 4. Information Integration Module**: When a patient response is returned to the Expert system, the information integration module aggregates all gathered patient information up to this point to update the understanding of the patient condition. This step simply appends a question-answer pair to the end of an existing conversation log, which will then be passed to the Abstention Module.

**Step 5. Decision Making Module**: When enough evidence is gathered, the Expert system leverages integrated patient information and medical knowledge to provide an accurate answer to the question. The input to this module is previous reasoning steps, the MCQ, and the gathered patient information, and the output is the chosen option. Exact prompts for all above sections are in Appendix B.

### 2.2.2 Expert System Variants with Different Abstention Strategies

One component of active information-seeking is the ability to decide *when* to ask questions, which we operationalize with the Abstention Module to either *ask* or *answer* at each turn. Abstention reduces LLM hallucinations in low-confidence scenarios (Umapathi et al., 2023; Rawte et al., 2023) and mitigates misleading or insufficiently substantiated conclusions (Feng et al., 2024). We develop the following variants of the Abstention Module via different instructions to the LLM to probe its confidence in whether their parametric knowledge is sufficient to reliably answer the MCQ. Exact prompts are in Appendix B.2.

**0. BASIC:** As a baseline, the model is asked to implicitly indicate its abstain decision by either generating an atomic question or producing an answer to the MCQ.

**1. Numerical:** To get an explicit understanding of the model's confidence, we first prompt the model to generate a numerical confidence score between 0 and 1 following (Tian et al., 2023). Then, an arbitrary threshold is set to either proceed with a final answer or ask a question.

2. **Binary:** Previous work has shown that LLMs struggle at producing numerical confidence scores (Srivastava et al., 2022; Geng et al., 2023). To address this, the Binary variant enables a simple classification of whether enough information is present. This setup simplifies the decision process, but may lack the nuanced understanding of confidence levels.

3. **Scale:** Binary classification does not provide granularity where the decision is ambiguous. Scale abstention solves this issue by combining direct quantification with a manageable set of discrete, interpretable options. The model is given definitions of confidence levels on a 5-point Likert scale (e.g., `"Very Confident"`, `"Somewhat Confident"`), and is asked to select a rating to express its confidence. An arbitrary threshold is set to either proceed with a final answer or ask a question.

4. **Rationale Generation (RG):** Model performance is shown to improve when prompted to generate a reasoning chain about the decision process (Wei et al., 2022; Marasović et al., 2021). This gives the model a longer context window for reasoning, allowing the final decision to be conditioned on previous generations. We attempt to generalize this finding to the more complex interactive medical information-seeking setup by applying it to Numerical, Binary and Scale abstention prompts.

5. **Self-Consistency (SC).** To further improve the Expert system's abstaining decision, we apply Self-Consistency to the above variants. Self-consistency repeatedly prompts the LLM $n$ times and take the average (Numerical and Scale) or the mode (Binary) of the output as the final output, and is shown to improve model performance (Wang et al., 2022).

## 3 Experiments

We conduct experiments to validate each component of MEDIQ. First, we evaluate the Patient system with *factuality* and *relevance* metrics (§ 3.1). Then, we establish the correlation between information availability and accuracy by studying model performance with varying levels of input information (§ 3.2.1). Finally, we improve the information-seeking ability of LLMs under MEDIQ (§ 3.2.2).

**Evaluation Dataset** We convert MEDQA (CC-BY 4.0) (Jin et al., 2021) and CRAFT-MD (CC-BY 4.0) (Johri et al., 2023, 2024) into an interactive setup for our experiments. MEDQA is a standard benchmark for medical question answering with 10178/1272/1273 train/dev/test samples. Each sample contains a paragraph of patient record ending with a multiple choice question. CRAFT-MD contains 140 dermatology patient records in a similar format, among which 100 are collected from an online question bank and 40 are created by expert clinicians. We parse each patient record into age, gender, the chief complaint (primary reason for the clinical visit), and additional evidence. Only the age, gender, and chief complaint are presented to the Expert system, from which it is expected to elicit missing information. The resulting tasks are called iMEDQA and iCRAFT-MD, respectively. See Appendix C for detail.

### 3.1 Patient System Reliability Evaluation

We automate the evaluation of patient responses with factuality score and relevance score for the ease of scalability, and conduct manual annotations to validate our metrics (Appendix A.4).

*Factuality Score* measures whether the Patient system's response is consistent with the patient record. Each Patient response is first decomposed into a list of atomic statements, then we compute the percentage of atomic statements that are supported by the information in the patient record. The factuality score is the percent of supported statements averaged over all patients.

*Relevance Score* measures whether the Patient system's response answers the Expert's question. Since there is no oracle data on the correct answer for Expert follow-up questions, we construct a synthetic parallel evaluation dataset of questions and responses to evaluate the relevance of Patient responses: given a patient record decomposed into atomic statements, we rephrase each statement into an atomic question, for which the statement is the ground truth answer. Then, the Patient system produces a response using the patient record and the generated atomic question. The average embedding semantic similarity between the generated response and the ground truth statement over the evaluation dataset is the resulting relevance score. See Appendix A.1 for more detail.

**Setup** We use GPT-3.5 as the base LLM for all three variants (Direct, Instruct, and Fact-Select) and compare the factuality and relevance scores. For factuality, we sample 1272 patient cases from MEDIQ interactions with follow-up questions generated by different Expert systems so the Patient

| Full | Initial | None |
|---|---|---|
| A 40-year-old woman presents with difficulty falling asleep, diminished appetite, and tiredness. She has grown increasingly irritable and hopeless [...] diminished concentration, [...] lost 8.8 lb [...] drinks a glass of wine instead of eating [...] What is the best treatment for this patient? | A 40-year-old woman presents with difficulty falling asleep, diminished appetite, and tiredness. What is the best treatment for this patient? | What is the best treatment for this patient? |
| (A) Diazepam  (B) Paroxetine  (C) Zolpidem  (D) Trazodone | Initial Info Details Question | |

**Figure 4:** Non-interactive Expert system evaluation at various information availability levels. The question and options are provided to the Expert model in all three settings.

system sees diverse Expert questions and compute the average across all generated questions. For relevance, we use all 1272 patient records from the development set of MEDQA.

### 3.2 Expert System Experiments

#### 3.2.1 Benchmarking Existing LLMs in Incomplete Information Scenarios

We evaluate the performance of non-interactive Expert systems with varying information availability levels to observe the relationship between information availability and accuracy and to establish baselines. The baselines are evaluated at three initial information availability levels (Figure 4): **Full**, **Initial**, and **None**. The **Full** setup is equivalent to the standard QA task, wherein all patient information is provided to the Expert system in the beginning; **Initial** only discloses the gender, age, and the chief complaint that leads to the clinical visit (e.g. fever, headache, etc.); **None** provides no patient information but only the MCQ to the Expert system.

#### 3.2.2 Interactive Expert Systems

**Expert Variants**   Without explicitly providing the option to ask follow-up questions, vanilla LLMs always answer with incomplete information and *never* ask for additional evidence. Therefore, we establish a question-asking Expert system baseline—**BASIC**—by prompting the LLM to either ask a question or make a decision at each turn. To study abstention, we combine Numerical, Binary, and Scale abstention with rationale generation and self-consistency techniques described in § 2.2.2.

**Expert System Setup**   We evaluate Llama-3-Instruct (8B, 70B), GPT-3.5, and GPT-4 on iMEDQA and iCRAFT-MD for both the non-interactive and interactive settings. Analysis and ablations use GPT-3.5 results on iMEDQA only. Details on model version and compute are in Appendix C.

**Expert System Evaluation Metric**   An ideal Expert system should be able to ask informative questions that allow it to arrive at accurate medical decisions efficiently. Since it is not trivial to measure the quality of medical information-seeking questions, we use the efficiency of the interaction (number of questions) and accuracy of the solution as proxies to evaluate the clinical reasoning capabilities. Accuracy is strongly dependent on the amount of information available to the model (§ 3.2.1), so higher accuracy is correlated with stronger information-seeking ability of the LLM.

## 4 Results

### 4.1 How reliable is the MEDIQ Patient system?

Our results in Table 1 show that both the **Direct** and **Instruct** settings struggle with factuality. Qualitative analysis revealed that since the Direct setting did not receive any instructions on *how* to respond to the follow-up question, it sometimes responds with "Yes" or "No" instead of the atomic statements that contain the requested information. In the Instruct setting, the Patient system sometimes provide inferences instead of reciting the facts from the patient record. Some example

| Model | Factuality | Relevance |
|---|---|---|
| **Direct** | 55.9 | 75.5 |
| **Instruct** | 62.8 | 78.6 |
| **Fact-Select** | **89.1** | **79.9** |

**Table 1:** Patient system reliability.

failure cases are shown in Appendix A.3. On the other hand, the **Fact-Select** setting which generates the responses in a more controlled environment increases factuality by 0.33 points and relevance by 0.04 points. Overall, these results suggest that *using atomic facts as units of information significantly reduces hallucination*, improving the reliability of the Patient system in providing accurate and relevant responses to expert questions. We use the Fact-Select setting for the Patient system in all subsequent experiments and shift our focus to evaluate the Expert variants introduced in § 3.2.

| Task | Model | Non-Interactive | | | Interactive | |
|------|-------|------|---------|------|-------|------|
| | | **Full** | **Initial** | **None** | **BASIC** | **BEST** |
| iMEDQA | **Llama-3-8b** | 68.1±1.3 | 52.0±1.4 | 40.4±1.4 | 33.0±1.3 | 45.8±1.4 |
| | **Llama-3-70b** | 84.7±1.0 | 58.5±1.4 | 46.3±1.4 | 55.1±1.3 | **60.9**±1.4 |
| | **GPT-3.5** | 55.8±1.4 | 45.6±1.4 | 36.7±1.4 | 42.2±1.3 | **50.2**±1.4 |
| | **GPT-4** | 79.7±1.1 | 54.5±1.4 | 42.2±1.4 | **55.4**±1.4 | **66.1**±1.3 |
| iCRAFT-MD | **Llama-3-8b** | 76.4±3.6 | 51.4±4.2 | 29.3±3.8 | 42.9±4.2 | 50.0±4.2 |
| | **Llama-3-70b** | 82.1±3.2 | 60.7±4.1 | 52.9±4.2 | **62.1**±4.1 | **72.1**±3.8 |
| | **GPT-3.5** | 82.1±3.2 | 53.6±4.2 | 29.3±3.8 | 45.0±4.2 | **59.3**±4.2 |
| | **GPT-4** | 91.4±2.4 | 67.9±3.9 | 43.6±3.7 | **73.6**±3.7 | **84.3**±3.1 |

**Table 2:** Accuracy at varying information availabilities. BASIC gives LLM the option to ask questions: with the same starting information, BASIC performance degrades from non-interactive Initial. Bold entries surpass non-interactive Initial, but there is still a gap between Full (complete information upper bound) and interactive BEST.

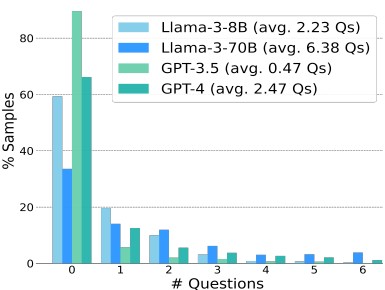

**Figure 5:** Frequency of conversation lengths in the BASIC setting. Most models don't tend to ask follow-up questions.

## 4.2 How do existing Non-Interactive LLMs perform with Limited Information?

As shown in Table 2, with decreasing amounts of patient information provided to the model, there is a pronounced drop in performance from the **Full** to **Initial** to **None** information availability levels. Shifting our attention to the **BASIC** interactive setup, the final accuracy is even lower than its non-interactive counterpart (Initial) with the same initial information (a average of 11.310.3%relative drop). We analyze performance sensitivity to prompt variations to ensure a fair comparison and report results from additional LLMs in Appendix E.

Figure 5 shows the number of follow-up questions asked by the LLMs in the BASIC interactive setup. For majority of the samples, *no* model chooses to ask any questions, showing the lack of ability in LLMs to proactively identify and elicit missing information. Within each LLM family (Llama/GPT), there is a correlation between model size, number of questions asked and accuracy. Overall, these results show a significant gap between model performance in idealized settings and realistic, information-limited scenarios. None of the examined models excel at proactive information seeking in an interactive environment, suggesting that it is nontrivial to integrate information gathered from continuous interactions. Despite having some medical knowledge encoded during pretraining, LLMs struggle to compensate for the absence of detailed patient information, highlighting the need for advanced proactive information-seeking abilities in medical LLM applications.

## 4.3 How much of the performance gap can be closed by asking questions?

In Figure 6, We present a summary of the information-seeking ability of MEDIQ Expert models with different abstain strategies by reporting the *accuracy* and *number of questions* (conversation efficiency). Recall that both the Numerical and Scale abstention methods require setting a confidence threshold, above which the Expert system will proceed to the final answer. We do a grid search for the threshold hyperparameter in Appendix D and report the best performance for each setting. Integrating a dedicated Abstention Module significantly enhances performance over the BASIC setup which directly prompts for follow-up questions or diagnoses. As the abstain strategies improve – by expressing confidence on a scale, verbal reasoning, and adding self-consistency – the expert model is able to better gauge the (lack of) patient information and continue the conversation by asking more questions and thereby improving the final accuracy.

Base abstention methods (Numerical, Binary, Scale) show little variance in effectiveness until combined with rationale generation, which consistently boosts performance across strategies, as supported by previous studies (Marasović et al., 2021; Wei et al., 2022; Feng et al., 2024). Notably, self-consistency alone *degrades* performance unless paired with rationale generation. Overall, the Scale Abstention (1-5 confidence rating) with Rationale Generation and a Self-Consistency factor of 3 achieves the best performance. Overall, Scale Abstention (1-5 confidence rating) with rationale generation and a self-consistency factor of 3 achieves the best performance, outperforming the BASIC interactive setup by 22.3% and the non-interactive Initial setup by 12.1%. In information-scarce scenarios, models tend to resort to the most common option instead of specializing to the patient, and interaction enhances specialization (Appendix F).

This pattern is generalizable across different LLMs as shown in the BEST column of Table 2. Note that model size plays a big factor in the performance of the interactive setting—models larger than 70B surpass the non-interactive Initial setup with the best abstention, but smaller models still struggle.

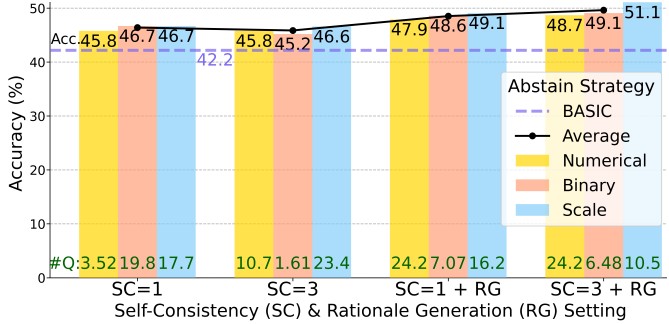 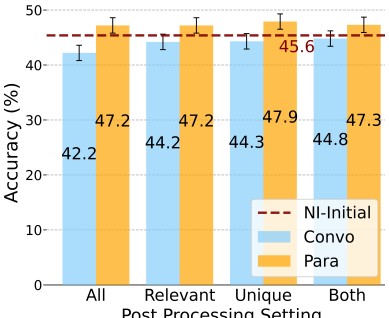

**Figure 6:** Accuracy of different abstaining strategies (with # follow-up questions noted in green). Self-consistency tend to improve performance only when combined with rationale generation.

**Figure 7:** BASIC error analysis. Removing irrelevant info and reformatting conversation improve accuracy.

While informing LLMs *when* to ask questions through abstention helps improve their decision-making with limited information, our best MEDIQ Expert system (Scale+RG+SC) still only closes 51.2% of the gap between the *Non-Interactive Initial* and the *Full* information scenarios. This indicates plenty of room for improvement to further enhance the information-seeking ability of LLMs.

# 5 Analysis

In this section, we further analyze the factors that impact the performance of the interactive Expert system. Since we observe similar trends across models and datasets, all analysis will be performed on the iMedQA dataset with GPT-3.5 due to cost and computation constraints.

## 5.1 Why does the BASIC interactive setup fail to perform clinical reasoning?

Recall from § 4.2 that there is a striking 11.3% relative drop in accuracy from BASIC to the *non-interactive* Initial information setting (NI-Initial) across all benchmarked LLMs (7.43% for GPT-3.5 on iMEDQA). In this section, we analyze failure modes of the BASIC system, where the Expert is simply given the option to ask follow-up questions, to understand the performance drop. We show that the ability to ignore irrelevant context and extract useful information from conversation format affects model performance.

**Irrelevant Context** There are two types of irrelevant context on model performance: *unanswerable* and *repeated* questions. As MEDIQ allows the Expert to ask any open-ended questions to the Patient to elicit information, some questions cannot be answered using the patient record. We filter out these unanswerable question-response pairs, keeping only record-based questions and responses to assess the effect of ignoring irrelevant questions (**Relevant**). Secondly, although the Expert is instructed to not repeat any questions, upon inspection of the interaction history, many questions are repetitive, especially when the answer is not in the patient record. We hypothesize that the repetition shifts the model's attention to certain questions and thus hinders the performance. We remove repetitive questions and only keep the unique questions (using fuzzy lexical matching) to verify this hypothesis (**Unique**). Finally, we remove both unanswerable and repeated questions (**Both**).

**Conversation Format** We further hypothesize that the dialogue format, different from the typical document format seen during LLM pre-training, also affects performance. To control for this, we convert the conversation format into paragraph format by discarding the Expert questions and only keeping the patient response for answerable questions, and rewriting the unanswerable questions into statements (e.g., `The patient's vaccine record is unavailable.`) for each setting above.

As shown in Figure 7, Relevant and Unique both improve performance by 2 percentage points (pp), but the combined effect is indistinguishable from using either filter, which might be due to the fact that unanswerable questions tend to be repeated. Converting the conversations into paragraph format further improves the performance (Para). Removing repetitive questions and converting to paragraph format (Unique-Para) surpasses BASIC by 5.7pp and NI-Initial by 2.3pp. This shows that, when given the option to ask follow-up questions, the information-seeking ability of the Expert system does help make more informed and accurate conclusions, but the model suffers from not being able to learn from realistic clinical dialogues.

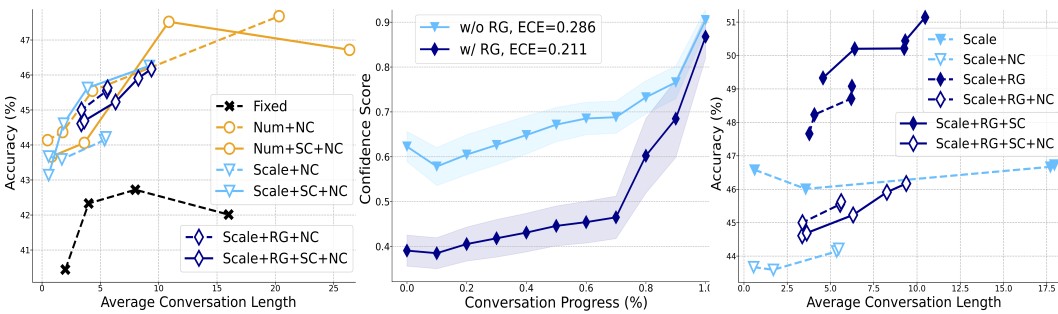

**Figure 8:** Effect of the abstention module on Expert system performance (efficiency & accuracy). (a) Accuracy over the number questions when the abstention response is *not* provided to the question generator **c**ontext (NC) to isolate the effect of confidence thresholds on model performance. Each line is a different abstain method with increasing confidence thresholds at each point. **Accuracy increases with # Qs as threshold increases.** (b) Expert confidence estimation throughout the interaction with and without rationale generation (RG), averaged over self-consistency levels 1, 3, and 5. The expected calibration error (ECE) is shown in the legend. **RG leads to more conservative and accurate confidence estimates.** (c) Accuracy of Expert system variants with abstention response available (solid marker) vs. not available (hollow marker) to the question generator module. **Including abstention response in question generator context improves accuracy, and is amplified with rationale generation (RG) and self-consistency (SC).**

## 5.2 How does abstention improve Expert system performance?

As we saw in § 4.3, the Abstention Module is effective in improving the performance of the Expert system. We decompose the clinical reasoning process of the Expert into deciding *when* to ask questions and *what* question to ask, and show that both contribute to performance gains.

**Abstain Threshold Affects Number of Questions and Accuracy** In the Expert systems described in § 2.2, the question generation module takes the abstention response as part of the context to produce a follow-up question. We now investigate the effect of the abstention decision by removing the abstention response from the question generation module, which allows us to control for the same question generator and only vary the abstention module. We plot the Expert performance across various abstention strategies and confidence thresholds in Figure 8(a). As the confidence threshold becomes higher (i.e., when the model becomes more cautious), the conversation length intuitively grows, and accuracy also increases. Notice that in the **Fixed** setting, the abstention decision is *not* specialized to the current patient, but decided solely by the number of questions asked; this results in a drop in accuracy when the number of questions becomes too high. While all methods benefit from longer interactions, the accuracy still varies depending on the abstention strategy, suggesting that some abstention strategies are better at estimating model confidence.

**Rationale Generation Influences Confidence and Performance** To further investigate confidence estimation, Figure 8(b) plots the intermediate confidence scores of the Scale-abstention Experts with and without rationale generation (RG) throughout the interaction, averaged over all threshold and self-consistency levels. We observe that RG leads to more conservative confidence estimates and lower expected calibration error (2.1 with RG vs. 2.9 without RG), which indicate that it enables more accurate identification of knowledge gaps via reasoning (Figure 9). These findings suggest that better confidence estimation lead to better interactive Experts, encouraging future work to extend confidence calibration techniques to interactive settings (Xie et al., 2024; Yang et al., 2024). Secondly, Figure 8(c) shows that including the abstention module response—confidence estimation and rationale—in the context of the question generation module significantly improves performance. Together, the results suggest that rationale generation doesn't differ from its no-rationale counterparts in terms of knowing when to ask questions (Figure 8(a)), but contributes in asking better questions (Figure 8(c)). This process essentially extracts the model's internal medical knowledge to integrate into subsequent responses, from which we conjecture that rationale generation can be further improved by integrating external specialized domain knowledge (Feng et al., 2023) or collaborative decision making with other models and/or humans (Talebirad & Nadiri, 2023). Additionally, we show that the nature of the question (medical specialty and difficulty) impacts the Expert's interactive behavior in Appendix G.

| Abstention Module Response |
|---|
| **REASON:** The patient provided current medications but did not mention any known history of liver disease or allergies. |
| **DECISION:** Somewhat Unconfident. |
| Question Generation Module Response |
| **ATOMIC QUESTION:** Have you ever experienced any symptoms like yellowing of the skin or eyes, abdominal pain, or dark urine which might indicate a liver problem? |

**Figure 9:** Example of rationale generation helping identify knowledge gaps to ask better follow-up questions.

# 6   Related Work

**Medical Question Answering Systems**   Advancements in medical question answering (QA) systems have progressed from rule-based systems to LLM-powered agents. Notable medical QA benchmarks include MultiMedQA (Singhal et al., 2023a), which contains both multiple-choice and open-ended questions collected from various sources. To customize or adapt a general-purpose LLM to the medical domain, prior work often finetune the model on medical knowledge data such as PubMed (Bolton et al., 2022; Yasunaga et al., 2022; Wu et al., 2023a; Singhal et al., 2023a,b), or more recently on conversational medical datasets (Yunxiang et al., 2023; Han et al., 2023). Kim et al. (2024) further improves model performance on complex medical questions by dynamically forming multi-agent collaboration structures. Despite their proficiency in direct answer retrieval, the proactive information-seeking capability is not something these models are inherently designed to do. Our proposed methodological framework, MEDIQ, is designed to work as an overlay to these domain-specific models, enhancing them with the capability to actively seek additional information in a structured and clinically relevant manner.

**Interactive Models and Agents**   Interactive conversational models extend beyond the standard QA framework by engaging in a dialogue such as customer support and negotiation (Singh & Beniwal, 2022; Chakrabarti & Luger, 2015; He et al., 2018; Abdelnabi et al., 2023; JU et al., 2024), where iterative information gathering is crucial. Li et al. (2023) and Andukuri et al. (2024) attempt to use LLMs to elicit more information-rich human preference examples in everyday tasks. However, the application of these models in the medical domain remains limited (Li et al., 2021). Wu et al. (2023b) attempts to evaluate general-purpose LLMs and chain-of-thought reasoning on DDXPlus (Fansi Tchango et al., 2022), a rule-based synethtic patient dataset. Hu et al. (2024) navigates the information-seeking scenario as a search problem by developing a reward model guided by uncertainty and includes medical diagnosis as one of the tasks, but if limited to binary questions. Johri et al. (2023) observed a similar phenomenon where LLM-doctors cannot elicit complete patient information, but do not focus on improving the information-seeking ability. The system proposed by (Tu et al., 2024) performs a multitude of medical tasks but does not explore the crucial problem of abstention. Lastly, multi-agent and human-AI collaboration frameworks have shown impressive interactive performance (Zhou et al., 2024; Wu et al., 2024, 2023c; Deng et al., 2024; Lin et al., 2024), and can greatly benefit from our novel interactive abstention methods to seek additional information. Our work fills this gap via providing a benchmark, a dataset, and a framework to comprehensively studying information-seeking abilities in clinical decision-making, and most importantly, opens the door for future endeavors in this direction.

# 7   Conclusion

In this paper, we identify a significant gap in current LLMs' capability to ask questions and proactively seek information in settings where personalization, precision, and reliability are critical. We propose a paradigm shift to interactive benchmarks by simulating more realistic clinical interactions where only partial information is provided initially by introducing MEDIQ. MEDIQ provides a benchmark to the community to evaluate the question-asking ability of LLMs, contributing towards developing reliable models. We showed that SOTA LLMs like Llama-3 and GPT-4 struggle to gather necessary information for accurate medical decisions. To address this problem, we presented MEDIQ-Expert— a novel Expert system with improved confidence judgment and medical expertise, substantially improving clinical reasoning performance. MEDIQ operationalizes interactive and explicit clinical reasoning processes, with added interpretability in the reasoning flow of language models and decision making. We encourage future research to extend MEDIQ to more diverse Patient systems, expand medical knowledge integration, and customize the interactions to better serve the healthcare community.

## Limitations

One limitation is the scarcity of datasets that contain detailed patient information sufficient for a medical diagnosis which, to the best of our knowledge, was only met by MEDQA and CRAFT-MD. The majority of available medical datasets are designed to test models' medical domain knowledge. Second, the Patient system in our benchmark relies on a paid API; future work should establish an open-source Patient. Lastly, our evaluation framework, while designed to be more realistic, is still limited to the multiple-choice format. However, the flexibility of MEDIQ allows easy extensions into open-ended settings with appropriate datasets and well-defined conversation-level metric. Future work can focus on collecting a rich dataset in open-ended medical consultations and expanding the MEDIQ framework.

## Ethics Statement

Along with many potential benefits of an ideal future variant of the MEDIQ framework (e.g., providing reliable and personalized medical consultation when access to medical experts is unavailable, or assisting medical experts in initial collection of information), it is important to emphasize multiple risks associated with mis-use of this framework.

First, MEDIQ is a carefully designed initial prototype, it is not meant to be deployed to interact with users; its intended use is to provide an experimental framework to test clinical reasoning abilities of LLMs which are currently extremely limited.

MEDIQ built on top of closed-source LLMs runs the risk of leaking confidential medical information, violating patient privacy. Future research expanding MEDIQ to new medical datasets should be aware of these risks, resorting to local securely stored LLMs or to reliable data anonymization methods.

There are many sources of potential biases in the framework, including social and cultural biases in LLMs, in the datasets, and possibly in prompts for LLM interactions and abstention that we designed. While outside the scope of this paper, in addition to utility metrics we proposed, future research could incorporate fairness-oriented evaluations, e.g., breaking down the evaluation by user demographics.

If a similar framework is used in real-world applications, users and clinicians should be trained to prevent the over-reliance on technology that is liable to make mistakes, and to understand its privacy and fairness risks.

## Acknowledgements

We gratefully acknowledge support from the University of Washington Population Health Initiative. SSL, SF, and YT are supported by the National Science Foundation under CAREER Grant No. IIS2142739, NSF grants No. IIS2125201, IIS2203097, and gift funding from Google, MSR, and OpenAI. PWK is supported by the Singapore National Research Foundation and the National AI Group in the Singapore Ministry of Digital Development and Innovation under the AI Visiting Professorship Programme (award number AIVP-2024-001). EP was supported by a Google Research Scholar award, Optum, NSF CAREER #2142419, a CIFAR Azrieli Global scholarship, a gift to the LinkedIn-Cornell Bowers CIS Strategic Partnership, and the Abby Joseph Cohen Faculty Fund.

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

# A  Patient System

## A.1  Evaluation Metrics

The Patient System is responsible for answering follow-up questions from the Expert System to provide the inquired patient information. When answering the follow-up questions, the Patient System has access to 1) the full patient context and 2) the expert question, and is instructed to produce a factual answer grounded in the context information and make no inferences. To ensure a reliable information-seeking process, we evaluate the Patient System along two axes: **Factuality Score** and **Relevance Score**. The basis of both the factuality and relevance evaluation relies on atomic facts from the patient record as units of information, which we generate by prompting GPT-4 following Min et al. (2023).

Factuality score measures whether the responses produced by the patient model are factual with respect to the given patient context information:

$$\text{factuality}(\mathcal{R}, \mathcal{C}) = \frac{1}{|R|} \sum_{r_i \in R} \frac{\sum_{j=1}^{|r_i|} \mathcal{I}(r_i^j, \mathcal{C})}{|r_i|}, \tag{1}$$

where $\mathcal{R}$ is the responses from the Patient System on the patient case $\mathcal{C}$; the size of $\mathcal{R}$ depends on the number of deduplicated Expert questions on the patient $\mathcal{C}$. $\mathcal{I}$ is the indicator function of whether atomic fact $r_i^j$ from the response $r_i$ is factually consistent ($\approx$) with any statement $c$ in $\mathcal{C}$:

$$\mathcal{I}(r_i^j, \mathcal{C}) = \begin{cases} 1, & \text{if } \exists\, c \in \mathcal{C} \text{ s.t. } r_i^j \approx c \\ 0, & \text{otherwise}, \end{cases} \tag{2}$$

Factual consistency of the response and any statement $c$ in $\mathcal{C}$ ($r_i^j \approx c$) is determined using semantic similarity for non-first person response patient variants and GPT-4 binary classification for first-person responses, since the atomic facts $c_i$ are written in third person. Specifically, $r_i^j \approx c$ in $\mathcal{I} = 1$ true if the semantic similarity, calculated by the cosine similarity of the SentenceTransformer `stsb-roberta-large` embeddings, is greater or equal to $0.8$ (or if the two sentences are determined to be consistent by a GPT-4 judge in the first-person response case), and vice versa.

**Relevance Score** measures whether the generated patient response answers the expert question. Defined similarly to the hallucination rate:

$$\text{relevance} = \frac{1}{|R|} \sum_{r_i \in \mathcal{R}} \frac{\sum_{j=1}^{|r_i|} \mathcal{I}(r_i^j, \mathcal{F}_i)}{|r_i|}, \tag{3}$$

where $\mathcal{F}_i$ is the ground truth statements from the patient context that answer the same expert question that $r_i$ is trying to answer.

## A.2  Patient System Variants

### A.2.1  Patient System Variant: Direct

> **Prompt to LLM**
>
> ```
> Answer the question with the given context:
> Context:  "She says that, despite going to bed early at night, she is unable
> to fall asleep.  She denies feeling anxious or having disturbing thoughts
> while in bed.  Even when she manages to fall asleep, she wakes up early
> in the morning and is unable to fall back asleep.  She says she has grown
> increasingly irritable and feels increasingly hopeless, and her concentration
> and interest at work have diminished.  The patient denies thoughts of suicide
> or death.  Because of her diminished appetite, she has lost 4 kg (8.8 lb)
> in the last few weeks and has started drinking a glass of wine every night
> instead of eating dinner.  She has no significant past medical history and is
> not on any medications."
> Question:  "What time do you usually go to bed at night?"
> ```

> **Example LLM Output**
>
> The patient goes to bed early at night despite experiencing difficulty falling asleep.

### A.2.2 Patient System Variant: Instruct

> **Patient Response Generation Prompt – Instruct**
>
> You are a truthful assistant that understands the patient's information, and you are trying to answer questions from a medical doctor about the patient.
>
> Below is a context paragraph describing the patient and their conditions: "She says that, despite going to bed early at night, she is unable to fall asleep. She denies feeling anxious or having disturbing thoughts while in bed. Even when she manages to fall asleep, she wakes up early in the morning and is unable to fall back asleep. She says she has grown increasingly irritable and feels increasingly hopeless, and her concentration and interest at work have diminished. The patient denies thoughts of suicide or death. Because of her diminished appetite, she has lost 4 kg (8.8 lb) in the last few weeks and has started drinking a glass of wine every night instead of eating dinner. She has no significant past medical history and is not on any medications."
>
> Question from the doctor: "What time do you usually go to bed at night?"
>
> Use the context paragraph to answer the doctor question. If the paragraph does not answers the question, simply say "The patient cannot answer this question, please do not ask this question again." Answer only what the question asks for. Do not provide any analysis, inference, or implications. Respond with a straightforward answer to the question ONLY and NOTHING ELSE.

> **Example LLM Output**
>
> Despite going to bed early at night, the patient is unable to fall asleep.

### A.2.3 Patient System Variants: Fact-Select

> **Atomic fact decomposition Prompt**
>
> You are a truthful medical assistant that understands the patient's information.
>
> Break the following patient information into a list of independent atomic facts, with one piece of information in each statement. Each fact should only include the smallest unit of information, but should be self-contained. "She says that, despite going to bed early at night, she is unable to fall asleep. She denies feeling anxious or having disturbing thoughts while in bed. Even when she manages to fall asleep, she wakes up early in the morning and is unable to fall back asleep. She says she has grown increasingly irritable and feels increasingly hopeless, and her concentration and interest

Response with the list of atomic facts and nothing else, prepend each fact by
an index starting from 1.  No sub-list allowed.

**Example LLM Output**

1.Patient goes to bed early at night but is unable to fall asleep.
2.Patient denies feeling anxious or having disturbing thoughts while in bed.
3.Patient wakes up early in the morning and is unable to fall back asleep.
4.Patient has grown increasingly irritable and feels increasingly hopeless.
5.Patient's concentration and interest at work have diminished.
6.Patient denies thoughts of suicide or death.
7.Patient has lost 4 kg (8.8 lb) in the last few weeks.
8.Patient started drinking a glass of wine every night instead of eating
dinner.
9.Patient has no significant past medical history.
10.Patient is not on any medications."

**Patient Response Generation Prompt – Fact-Select**

You are a truthful assistant that understands the patient's information, and
you are trying to answer questions from a medical doctor about the patient.

Below is a list of factual statements about the patient:
1.Patient goes to bed early at night but is unable to fall asleep.
2.Patient denies feeling anxious or having disturbing thoughts while in bed.
3.Patient wakes up early in the morning and is unable to fall back asleep.
4.Patient has grown increasingly irritable and feels increasingly hopeless.
5.Patient's concentration and interest at work have diminished.
6.Patient denies thoughts of suicide or death.
7.Patient has lost 4 kg (8.8 lb) in the last few weeks.
8.Patient started drinking a glass of wine every night instead of eating
dinner.
9.Patient has no significant past medical history.
10.Patient is not on any medications.

Question from the doctor:  "What time do you usually go to bed at night?"

Which of the above atomic factual statements answer the question?  Select
at most two statements.  If no statement answers the question, simply say
"The patient cannot answer this question, please do not ask this question
again." Answer only what the question asks for.  Do not provide any analysis,
inference, or implications.  Respond by selecting all statements that answer
the question from above ONLY and NOTHING ELSE.

**Example LLM Output**

Patient goes to bed early at night but is unable to fall asleep.

### A.2.4 Additional Patient System Variant: Fact-FP

In addition to the three Patient System variants presented in the main paper, we have explored two other variants: First Person (Fact-FP) and Binary Fact Classification (Fact-Classify). Fact-FP breaks down the patient record into atomic facts, select the relevant facts that answers the follow-up question, and respond by converting the atomic statements into first person perspective. Atomic fact decomposition prompt is the same as in Appendix A.2.3.

---

**Patient Response Generation Prompt – Fact-FP**

You are a patient with a list of symptoms, and you task is to truthfully answer questions from a medical doctor.

Below is a list of atomic facts about you, use ONLY the information in this list and answer the doctor's question.
1.Patient goes to bed early at night but is unable to fall asleep.
2.Patient denies feeling anxious or having disturbing thoughts while in bed.
3.Patient wakes up early in the morning and is unable to fall back asleep.
4.Patient has grown increasingly irritable and feels increasingly hopeless.
5.Patient's concentration and interest at work have diminished.
6.Patient denies thoughts of suicide or death.
7.Patient has lost 4 kg (8.8 lb) in the last few weeks.
8.Patient started drinking a glass of wine every night instead of eating dinner.
9.Patient has no significant past medical history.
10.Patient is not on any medications.

Question from the doctor:  "What time do you usually go to bed at night?"

Which of the above atomic factual statements answer the question?  Select at most two statements.  If no statement answers the question, simply say "I cannot answer this question, please do not ask this question again." Do not provide any analysis, inference, or implications.  Respond by reciting the matching statements, then convert the selected statements into first person perspective as if you are the patient but keep the same information. Generate your answer in this format:

STATEMENTS:
FIRST PERSON:

---

**Example LLM Output**

STATEMENTS: "Patient goes to bed early at night but is unable to fall asleep."
FIRST PERSON: "I go to bed early at night, but I can't fall asleep."

---

### A.2.5 Additional Patient System Variant: Fact-Classify

Atomic fact decomposition prompt is the same as in Appendix A.2.3.
Patient response generation prompt: for each atomic fact, repeat the following prompting process to decide whether it answers the doctor question, and collect a yes/no answer from the model. The final output is the set of atomic facts with a yes response.

**Patient Response Generation Prompt – Fact-Classify**

Does the statement answer the question? Do not provide any analysis,
inference, or implications. Respond with either YES or NO and NOTHING ELSE.

**Example LLM Output**

YES

## A.3 Qualitative Patient System Failure Analysis

This section provides a detailed analysis of failure cases observed in the MEDIQ Patient system variants Direct and Instruct. These examples highlight limitations and areas for potential improvement in Patient response accuracy and relevance.

**Patient System Variant: Direct** There are two common failures cases observed by the Direct Patient variant. First, it tends to respond with binary yes/no answers. Although the response is factually correct, it lacks the detail required for a comprehensive medical assessment. First, this behavior does not mimic a typical, realistic patient. Second, detailed information about symptoms, their progression, and related factors is crucial to reaching a diagnosis. Simple affirmative or negative responses do not provide enough context or specifics for an expert to make an informed decision about the patient's condition. The conversation history, which is compiled to assist in final diagnostic decisions, becomes less useful if it is filled with uninformative responses.

> Example 1:
> Atomic Question: Have you experienced any nausea recently?
> Atomic Fact (ground truth): Nausea has subsided.
> Patient Response: Yes.
> Example 2:
> Atomic Question: Have you experienced similar episodes in the past?
> Atomic Fact (ground truth): The patient has had 2 similar episodes in the last 5 years.
> Patient Response: Yes.

The second common error by the Direct Patient system is hallucinations. In the example below, the Patient system's response includes a hallucinated fact—claiming the initiation of insulin therapy, which is not supported by the patient record. This type of error is particularly problematic because hallucinations introduce false information into the patient's medical history, which can lead to inappropriate medical decisions or interventions based on incorrect data. False information regarding medication can have direct negative effects on patient safety, which calls for the development of factually reliable Patient systems.

> Example 1:
> Atomic Question: Can you describe any recent changes to your medication? Especially anything related to your diabetes?
> Atomic Fact (ground truth): The patient is not on any medications.
> Patient Response: I started taking insulin last week for my diabetes.

**Patient System Variant: Instruct**    The response below by the Instruct Patient system erroneously includes instructive content meant for the healthcare provider rather than addressing the question directly from the patient's perspective. This indicates a potential confusion in role-play within the interactive MEDIQ setup when the conversation log is compiled.



Atomic Question: Have you noticed any unusual bruising on your body?
Atomic Fact (ground truth): Patient experiences easy bruising.
Patient Response:  Due to the easy bruising experienced by the patient and her family history of similar issues, it is important to ask the patient if she has noticed any unusual bruising on her body.



### A.4   Manual Evaluation of Fact-Select

In order to better understand the quality of the Patient System variant used in subsequent Expert evaluations, we perform a manual annotation of the quality of the responses. A simple preference-based annotation is performed, in which the annotators are provided with a patient record, a Expert follow-up question, and two responses generated by two different Patient systems and are asked to select the one that is more factually correct and does not omit information. The annotators are also instructed to reduce their rating on binary Yes/No answers due to formatting requirements of the framework, but they are free to select any option during annotation. We randomly select one question from each interaction that contains at least 3 follow-up questions, and generate responses with all three Patient systems. For each question, two responses are randomly selected and the annotator is asked to indicate which response is of higher quality. Win rate for each Patient system is calculated as the number of times responses from a system is selected over the number of appearances of responses from this system. There is a tie option for if neither or both responses are good, which we exclude when calculating the win rate. 500 questions are annotated and results are shown in Table 3. We can see that there is an obvious advantage of Fact-Select Patient system over the other systems.

|  | **Direct** | **Instruct** | **Fact-Select** |
|---|---|---|---|
| **Win Rate** | 36.11 | 37.36 | 63.77 |

**Table 3:** Win rates from manual evaluation of Patient variants.

## B   Expert Model Prompts

### B.1   Initial Assessment



**MEDIQ Expert – Initial Assessment Module Prompt**

You are a medical doctor answering real-world medical entrance exam questions. Based on your understanding of basic and clinical science, medical knowledge, and mechanisms underlying health, disease, patient care, and modes of therapy, answer the following multiple choice question.  Select one correct answer from A to D. Base your answer on the current and standard practices referenced in medical guidelines.

Task:  You will be asked to reason through the current patient's information.

A patient comes into the clinic presenting with some basic information:
"A 40-year-old woman presents with difficulty falling asleep, diminished appetite, and tiredness for the past 6 weeks."



```
Given the information from above, your task is to choose one of four options
that best answers the inquiry.
INQUIRY: "Which of the following is the best course of treatment in this
patient?"
OPTIONS: "A":  "Diazepam", "B":  "Paroxetine", "C":  "Zolpidem", "D":
"Trazodone"

Medical conditions are complex, so you should seek to understand their
situations across many features.  First, consider which medical specialty
is this patient's case; then, consider a list of necessary features a doctor
would need to make the right medical judgment; finally, consider whether all
necessary information is given in the conversation above.  Think step by step,
reason about the patient information, the inquiry, and the options.  DO NOT
provide the answer choice, keep your response to one paragraph.
```

Example LLM Output

Based on the patient's presentation of difficulty falling asleep, diminished appetite, and tiredness, the most appropriate course of treatment would be to consider prescribing a medication typically used for depression, like paroxetine or trazodone. These medications can help address symptoms of depression, which can manifest as changes in sleep patterns, appetite, and energy levels. Diazepam and zolpidem are not typically indicated for primary treatment of underlying depressive symptoms. Further evaluation, including a proper assessment for depression and consideration of counseling or therapy, may also be beneficial in the overall management of this patient.

## B.2   Abstention Module

For this section, all prompts are appended after the prompt and response from Appendix B.1 in a multi-turn conversation manner.

### B.2.1   BASIC Abstain

MEDIQ Expert – BASIC Abstention Module Prompt (appended to previous conversation)

```
Considering factors above, if you are confident to pick an option correctly
and factually, respond with the letter choice and NOTHING ELSE. Otherwise, if
you are not confident to pick an option and need more information, ask ONE
SPECIFIC ATOMIC QUESTION to the patient.  The question should be bite-sized,
and NOT ask for too much at once.  In this case, respond with the atomic
question and NOTHING ELSE.
```

Example LLM Output

"What time do you usually go to bed at night?"

### B.2.2   Binary Abstain

MEDIQ Expert – Binary Abstention Module Prompt (appended to previous conversation)

```
Considering factors above, are you confident to pick the correct option to
the inquiry factually using the conversation log?  Answer with YES or NO and
NOTHING ELSE.
```

> **Example LLM Output**
>
> YES

### B.2.3 Numerical Abstain

> **MEDIQ Expert – Numerical Abstention Module Prompt (appended to previous conversation)**
>
> ```
> Considering factors above, what is your confidence score to pick the correct
> option to the inquiry factually using the conversation log?  Answer with the
> probability as a float from 0.0 to 1.0 and NOTHING ELSE.
> ```

> **Example LLM Output**
>
> 0.7

### B.2.4 Scale Abstain

> **MEDIQ Expert – Scale Abstention Module Prompt (appended to previous conversation)**
>
> ```
> Considering factors above, how confident are you to pick the correct option
> to the problem factually using the conversation log?  Choose between the
> following ratings:
>
> "Very Confident" - The correct option is supported by all evidence, and there
> is enough evidence to eliminate the rest of the answers, so the option can be
> confirmed conclusively.
> "Somewhat Confident" - I have reasonably enough information to tell that the
> correct option is more likely than other options, more information is helpful
> to make a conclusive decision.
> "Neither Confident or Unconfident" - There are evident supporting the correct
> option, but further evidence is needed to be sure which one is the correct
> option.
> "Somewhat Unconfident" - There are evidence supporting more than one options,
> therefore more questions are needed to further distinguish the options.
> "Very Unconfident" - There are not enough evidence supporting any of the
> options, the likelihood of picking the correct option at this point is near
> random guessing.
>
> Answer in the following format:
>
> DECISION: chosen rating from the above list.
> ```

> **Example LLM Output**
>
> Somewhat Confident

### B.2.5 Rationale Generation

> **MEDIQ Expert – Rationale Generation Prompt (appended to previous conversation)**
>
> ```
> Considering factors above, are you confident to pick the correct option to
> the inquiry factually using the conversation log?  Answer in the following
> format:
>
> REASON: a one-sentence explanation of why you are or are not confident.
> DECISION: YES or NO.
> ```

> **Example LLM Output**
>
> REASON: "I am confident in selecting the correct option based on the patient's presentation of symptoms suggestive of depression, which would warrant treatment with medications like paroxetine or trazodone according to standard medical guidelines for managing depressive symptoms."
> DECISION: "YES"

## B.3 Follow-up Question Generation

> **MEDIQ Expert – Question Generation Module Prompt (appended to previous conversation)**
>
> ```
> If there are missing features that prevent you from picking a confident
> and factual answer to the inquiry, consider which features are not yet
> asked about in the conversation log; then, consider which missing feature
> is the most important to ask the patient in order to provide the most
> helpful information toward a correct medical decision.  You can ask about
> any relevant information about the patient's case, such as family history,
> tests and exams results, treatments already done, etc.  Consider what are
> the common questions asked in the specific subject relating to the patient's
> known symptoms, and what the best and most intuitive doctor would ask.  Ask
> ONE SPECIFIC ATOMIC QUESTION to address this feature.  The question should
> be bite-sized, and NOT ask for too much at once.  Make sure to NOT repeat any
> questions from the above conversation log.  Answer in the following format:
>
> ATOMIC QUESTION: the atomic question and NOTHING ELSE.
> ```

> **Example LLM Output**
>
> "What time do you usually go to bed at night?"

## B.4 Information Integration

None-LLM step. Rearrange the atomic question and patient response into the conversation log. Known patient information $\mathcal{K}$:

> ```
> A patient comes into the clinic presenting with some basic information:
> "A 40-year-old woman presents with difficulty falling asleep, diminished
> appetite, and tiredness for the past 6 weeks."
>
> Conversation log:
> Doctor Question:  "What time do you usually go to bed at night?"
> Patient Response:  "Patient goes to bed early at night but is unable to fall
> asleep."
> ```

### B.5 Decision Making

> **MEDIQ Expert – Decision Making Module Prompt (appended to previous conversation)**
>
> ```
> Assume that you already have enough information from the above
> question-answer pairs to answer the patient inquiry, use the above
> information to produce a factual conclusion.  Answer in the following format:
>
> FINAL CHOICE: correct letter choice and NOTHING ELSE.
> ```

> **Example LLM Output**
>
> D

## C   Model Version and Compute

**Model Specs and Datasets**   We use 4bit quantization for Llama-2-Chat-70B and Llama-3-Instruct-70B, and 8bit quantization for Llama-2-Chat-7B, Llama-2-Chat-13B, and Llama-3-Instruct-8B to reduce GPU usage and preserve model utility (Huang et al., 2024). For the OpenAI models, we use the `gpt-3.5-turbo-0125` version for GPT-3.5 and the `gpt-4-turbo-2024-04-09` version for GPT-4. Since no training is needed, we use the development set of iMEDQA with GPT-3.5 to improve the Expert system for faster iteration of the abstention methods. Then, we generalize our findings to other LLM and the entire set of iCRAFT-MD to test our best system.

**Hyperparameters**   For both the Patient and Expert systems, we use a temperature of 0.5 and $top\_p = 1$ for top p sampling. Another important hyperparameter is the confidence threshold in the Abstention Module that determines when the Expert should stop asking more questions, which we explore in the Analysis section (§ 5.2) using a grid search instead of arbitrarily setting a value.

### C.1   Computing Hardware

We use CPU only for the GPT-based experiments, one A40 GPU for the smaller Llama models (7B, 8B, & 13B), and two A40 GPUs for the 70B models. Time duration of the experiments for each model is roughly proportional the average conversation length (number of follow-up questions) of the interaction, whether rationale is generated, and the self-consistency factor. Experiment time ranges from 30 minutes for GPT-3.5-based Expert systems with no rationale generation and no self-consistency, to 7 days for Llama-2-Chat-70B-based Expert systems with rationale generation, self-consistency, and a high confidence threshold (i.e., more questions are asked).

### C.2   Statistical Significance Testing

Since it is expensive to perform repeated runs for all experiments, we approximate the confidence interval using a binomial distribution, where $p$ is the accuracy of the model and $n$ is the dataset size. Therefore, the standard deviation can be calculated as:

$$SD_B = \sqrt{\frac{p(1-p)}{n}} \qquad (4)$$

As a sanity check, we repeat the set of experiments in Table 2 with GPT-3.5 and iCRAFT-MD to calculate the random seed standard deviation ($SD_R$) over one trials. We report the Binomial standard deviation for trial #1 (#1–$SD_B$) and the random seed standard deviation ($SD_R$) over one

| Model | Non-Interactive | | | Interactive | |
| | **Full** | **Initial** | **None** | **BASIC** | **BEST** |
|---|---|---|---|---|---|
| #1 | 82.1 | 53.6 | 29.3 | 45.0 | **59.3** |
| #2 | 78.6 | 55.0 | 29.3 | 47.1 | **57.1** |
| #3 | 78.6 | 52.9 | 31.4 | 43.6 | **57.1** |
| #4 | 80.0 | 52.1 | 27.9 | 43.6 | **60.0** |
| #5 | 80.7 | 52.1 | 30.0 | 46.4 | **57.1** |
| #1–$SD_B$ | 3.38 | 4.22 | 3.86 | 4.21 | 4.17 |
| $SD_R$ | 1.51 | 1.19 | 1.30 | 1.63 | 1.39 |

**Table 4:** Repeated trials GPT-3.5 on iMEDQA to compare the Binomial SD over one run and random seed SD over five runs, showing that Binomial SD is a reasonable estimation of the confidence interval with limited data.

trials in Table 4 to show that the Binomial standard deviation is a reasonable estimation of the confidence interval when we only have one run available.

## D  Detailed Abstention Results

In order to determine the confidence threshold above which the model should proceed to the final answer, we perform a grid search over the iMedQA dataset. The results are shown in Figure 10. Generally, when the threshold becomes higher, the Expert system asks more questions, and the performance also increases. But as the number of questions becomes too high, the performance stagnates, which might be due to the fact that a lot of the questions will become irrelevant and/or repetitive.

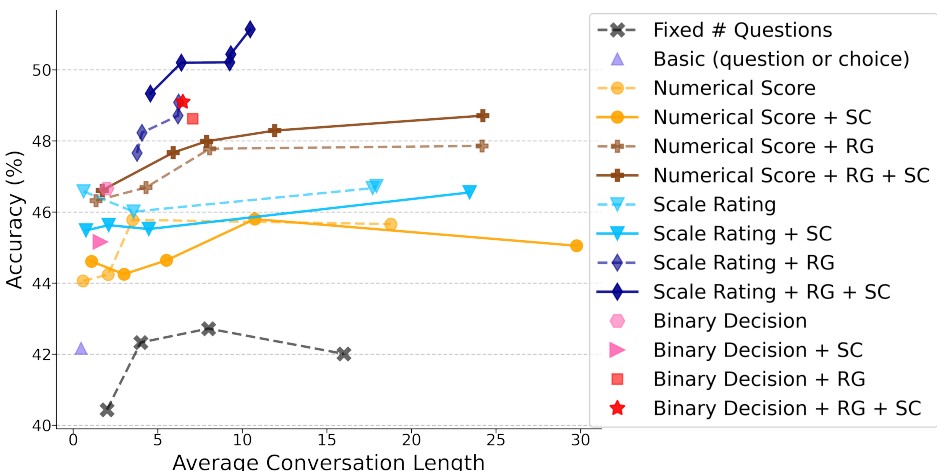

**Figure 10:** Performance of abstain strategies on iMEDQA. Each line is an abstain strategy with increasing confidence thresholds. Darker colors are results with rationale generation (RG); dashed lines are with self-consistency (SC). The BEST system (Scale+RG+SC,♦) significantly outperforms the BASIC baseline (▲).

## E  Results on Additional LLMs

We report the results of Experiment 3.2.1 on the Llama-2-Chat models below. Generally, we observe similar behavior in these models compared to models with higher utility (Llama-3, GPT-3.5, and GPT-4). However, due to the limited model capacity, results for a few settings are less reliable as they are close to at-chance performance (e.g., interactive BASIC for Llama-2-7B).

| Task | Model | Non-Interactive | | | Interactive | |
| | | Full | Initial | None | BASIC | BEST |
| --- | --- | --- | --- | --- | --- | --- |
| **iMEDQA** | **Llama-2-7b** | $30.8_{\pm1.3}$ | $26.3_{\pm1.2}$ | $26.8_{\pm1.2}$ | $27.8_{\pm1.3}$ | $\mathbf{31.9}_{\pm1.3}$ |
| | **Llama-2-13b** | $37.1_{\pm1.4}$ | $33.0_{\pm1.3}$ | $29.9_{\pm1.3}$ | $31.3_{\pm1.3}$ | $32.6_{\pm1.3}$ |
| | **Llama-2-70b** | $42.9_{\pm1.4}$ | $36.7_{\pm1.4}$ | $31.6_{\pm1.3}$ | $33.0_{\pm1.3}$ | $\mathbf{35.6}_{\pm1.3}$ |
| **iCRAFT-MD** | **Llama-2-7b** | $42.1_{\pm4.2}$ | $35.0_{\pm4.0}$ | $30.7_{\pm3.9}$ | $32.1_{\pm3.9}$ | $37.1_{\pm4.1}$ |
| | **Llama-2-13b** | $55.0_{\pm4.2}$ | $42.9_{\pm4.2}$ | $26.3_{\pm3.7}$ | $45.7_{\pm4.2}$ | $38.6_{\pm4.1}$ |
| | **Llama-2-70b** | $60.7_{\pm4.1}$ | $44.3_{\pm4.2}$ | $27.9_{\pm3.8}$ | $45.7_{\pm4.2}$ | $42.1_{\pm4.2}$ |

**Table 5:** Accuracy of Llama-2-Chat models with varying information availability. BASIC is LLM with basic prompting to ask additional questions. Bold results surpass the non-interactive Initial setup.

### E.1  Sensitivity to Prompt Variations

LLMs are sensitive to spurious features in the prompt Sclar et al. (2023). We first experiment on various single-turn baselines with prompt variations to evaluate the robustness of the LLaMa-2-chat models Touvron et al. (2023) on the prompt in order to finalize a prompt to standardize future runs. We explore three **system prompts**: Empty (default system prompt), Basic (You

are a helpful medical assistant), and the Meditron Prompt adopted from Chen et al. (2023). We also explore three **response prompts** (in the user message) for each combination of model size and system prompt: Answer Only (Respond with the correct option and nothing else), Rationale (Explain the rationale, then select the correct option), and Permutate (shuffling the option–answer pairs to remove potential dataset bias).

Results on the development set, depicted in Figure 6, show language models, particularly the Llama-2-Chat series, display consistency despite prompt variations. Performance modestly increases with rationale generation, and is unaffected by answer pair shuffling.

| Model Size | System Prompt | Response Prompt | Full Context | Basic Info | Question Only |
|---|---|---|---|---|---|
| **LLaMa-2 7b** | Empty | Answer Only | 33.04 | 32.10 | 31.94 |
| | | Rationale | 35.09 | 33.83 | 30.84 |
| | | Shuffle | 33.04 | 29.90 | 30.84 |
| | Basic | Answer Only | 32.49 | 33.12 | 28.48 |
| | Meditron | Answer Only | 30.81 | 26.83 | 26.28 |
| | | Rationale | 36.11 | 31.86 | 29.66 |
| | | Shuffle | 30.02 | 26.72 | 28.64 |
| **LLaMa-2 13b** | Empty | Answer Only | 38.00 | 36.03 | 31.63 |
| | | Shuffle | 37.14 | 34.23 | 30.53 |
| | Basic | Answer Only | 37.77 | 35.25 | 31.00 |
| | Meditron | Answer Only | 37.07 | 33.01 | 29.93 |
| | | Shuffle | 37.84 | 33.67 | 31.86 |
| **LLaMa-2 70b** | Empty | Answer Only | 43.87 | 36.66 | 33.88 |
| | Basic | Answer Only | 41.92 | 35.96 | 33.10 |
| | Meditron | Answer Only | 42.88 | 36.69 | 31.58 |
| | | Shuffle | 42.20 | 35.92 | 30.95 |
| **GPT-3.5** | Meditron | Answer Only | 53.42 | 43.12 | 35.72 |

**Table 6:** Prompt Variations & Single-Turn Baselines.

# F   How does prior knowledge influence model choice?

General-purposed LLMs tend to provide the most general answers. We hypothesize that this is because the model does not have enough information provided in the context, so it needs to resort to its parametric knowledge and chooses the most common option to maximize the likelihood of providing the correct answer. Therefore, when given more information in the context, the model should rely less on its parametric knowledge and customize to the patient. We quantitatively show this trend by obtaining the most common option according the Expert and calculate the *generality agreement*—percent agreement between the model choice and its belief of the most common option. Selecting the most common option regardless of context yields the correct response only 33.7% of the time, emphasizing the need for the model to focus on provided context rather than prior assumptions from its parametric knowledge. Results show that in setups with increasing information availability (**None**, **Initial**, **Full**), generality agreement shows an decreasing pattern: 50.0%, 40.2%, and 43.4%, respectively.

# G   How does the nature of the question influence model behavior?

In this section, we show that the nature of the question, such as medical specialty and difficulty, impacts the Expert's interactive behavior. Overall, interaction benefits more difficult questions and certain specialties such as ophthalmology. Figure 11 shows the impact of interactions on diagnostic accuracy across various medical specialties and demographics. Notably, Ophthalmology showed a significant improvement in accuracy from 18.2% to 45.5% after interaction (Figure 12), highlighting the model's potential in specialties with initially low accuracy. Similar trends are evident

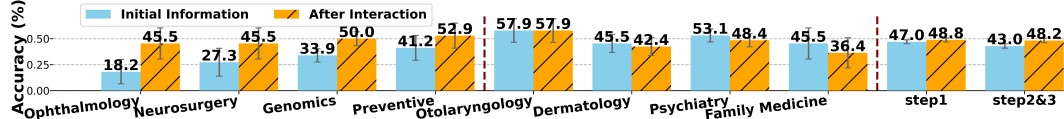

**Figure 11:** Impact of interactive system on diagnostic accuracy across medical specialties and demographics. Specialties benefiting the most (left) and least (middle) from interaction; improvement by difficulty (right).

in neurosurgery and genomics, suggesting that interactive information-seeking can enhance decision-making. However, interactions in specialties such as family medicine and psychiatry sometimes result in confusion. The effect of interaction on accuracy also extends across different difficulty levels of medical inquiries. For questions that are more challenging and clinically focused (Step 2 & 3), accuracy improved from 43.4% to 48.3%. Overall, this analysis highlights the importance of tailoring interactive diagnostic systems to specific specialties and types of questions for the maximal benefit.

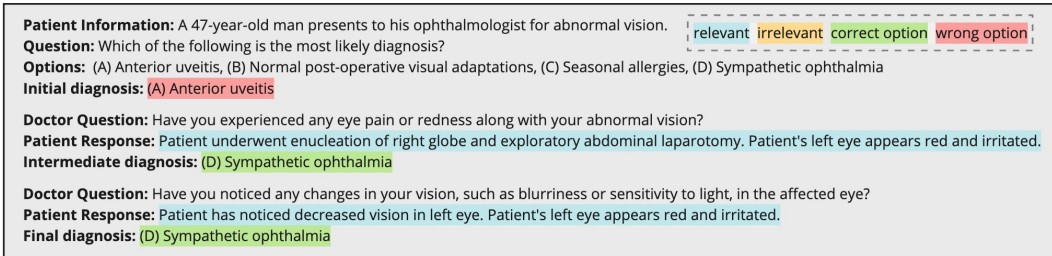

**Figure 12:** Example successful interaction in Ophthalmology.

