# OpenReview forum: "MediQ: Question-Asking LLMs and a Benchmark for Reliable Interactive Clinical Reasoning"
_NeurIPS.cc/2024/Conference — NeurIPS 2024 poster_

### Official Review · Reviewer_pkFC · 2024-06-30

**Soundness:** 3
**Presentation:** 3
**Contribution:** 3
**Rating:** 6
**Confidence:** 4

**Summary:**

This paper proposes a novel MEDIQ framework to simulate clinical interaction between doctor and patient. The proposed MEDIQ framework is capable of proactively asking questions and collecting information to make the diagnosis results. Also, the MEDIQ framework incorporates two agents—expert and patient, to formulate a conversation loop. With an abstaining module, the expert agent determines when to stop a conversation. The extensive experiments validate the design of the MEDIQ framework.

**Strengths:**

1.	The novel design of the expert and patient modules to formulate a medical conversation
2.	The design of the expert agent to proactively collect information to support the diagnosis decision
3.	The design of the patient agent to extract the information from the patient record and interact with the expert agent

**Weaknesses:**

1.	A large portion of the algorithm, for example, the abstention module and rationale generation module, rely on the output from ChatGPT, which makes the results less reproducible. Moreover, the outcome from ChatGPT, even though the authors claim they estimate model confidence, is still in a state of ‘black box’. The authors should develop some other methods to better quantify and evaluate the outputs, instead of simply relying on ChatGPT or other LLMs.
2.	The dataset is crafted using LLMs on the medical Q&A dataset. To better validate the performance of the proposed model, the authors should add experiments using the medical conversation dataset. In the context of real clinical conversation, the doctors may make a diagnosis decision not simply based on existing medical records, but also take symptoms, medical testing results, and other information into consideration. This information is not included in the medical Q&A dataset.

**Questions:**

1.	The authors use general LLMs to play as expert and patient agents. Will the performance be improved if the LLMs are finetuned for the medical domain?
2.	 How is the performance compared with Medical Q&A LLMs, such as MedPalm2?

**Limitations:**

1. Need to strengthen the explainability of the proposed framework, especially for the abstention module and rationale generation module.
2. Need to add medical conversation dataset for better evaluation.
3. Need to compare with medical Q&A LLMs.

---

> ### Author Rebuttal · Authors · 2024-08-07
>
> We appreciate the reviewer’s valuable feedback and insights. Thank you for highlighting the strengths of MediQ, including the novel design of the Patient and Expert two-agent conversation system, proactive information seeking feature in the Expert, and the robust Patient system to access patient records. We hope to address any questions below:
>
> > Need to strengthen the explainability of the proposed framework.
>
> We agree with the reviewer that ChatGPT is essentially a black-box, which is a shared concern in most LLM resarch. We ensure reproducibility by releasing all prompts, the static API versions (line 778-784), and also evaluating open-source models such as the Llama family. We also updated the Patient system with an open-source model (Llama-3) to reduce reliance on ChatGPT, and we will add the results to the camera ready version; the overall trends stay consistent.
>
> We use the **expected calibration error** to quantitatively evaluate the confidence estimators (line 352-354, Figure 8(b)). Our evaluation of each part of the pipeline component *do not* solely rely on ChatGPT or other LLMs. For example, factuality of the Patient system relies on semantic similarity and the overall Expert performance is evaluated with the diagnostic accuracy of MedQA and Craft-MD multiple choice questions.
>
> > Need to add medical conversation dataset for better evaluation.
>
> Most existing medical conversation datasets, such as ChatDoctor [1], are in the form of patients asking questions and doctors responding with answers in a single turn. We also considered KaMed [2], but since it lacks well-formulated multiple choice questions, it cannot be used for the current MediQ evaluation setup, where diagnostic accuracy is highlighted. Thus, given the scarce existing medical datasets, we did the best we could to operationalize the MediQ framework with MedQA and Craft-MD. Curating a *high-quality clinical interaction dataset* is a research direction that would greatly benefit research in proactive information seeking.
>
> Information such as **symptoms** and **medical testing results** are already in the datasets we use. As stated in the limitations section, one of the requirements of the datasets is to include sufficient contextual information about the patient. We present an example below.
>
> > ```A 5-year-old girl is brought to the emergency department by her mother because of multiple episodes of nausea and vomiting… She has been hospitalized 2 times during the past 6 months… Her immunizations are up-to-date… She appears emaciated. Her temperature is 36.8°C (98.8°F), pulse is 99/min, and blood pressure is 82/52 mm Hg. Examination shows dry mucous membranes. The lungs are clear to auscultation…```
>
> If the reviewer meant that when the information requested by the Expert is not present in the context, our current Patient system will refuse to answer since we assume that it will not aid in the diagnostic process. But we agree that in order to create an even more realistic scenario, we can have the Patient system answer out-of-scope questions by *creating more complete personas* and augmenting them with medically consistent details. We leave this for future work and will clarify in the updated version.
>
> > How is the performance compared with Medical Q&A LLMs, such as MedPalm2?
>
> We agree with the reviewer that medical LLMs can potentially outperform general purpose models. There are a few reasons why we did not include medical LLMs in our experiments.
>
> First, general-purpose LLMs such as GPT-4 can outperform domain-specific models on tasks such as MMLU and MedQA [4-7]. Second, most of the medical LLMs were fine-tuned on the training set of MedQA [5,6], which might give it shortcuts in predicitng the letter choice rather than perform explicit clinical reasoning as what the MediQ framework is designed to accomplish. Third, medical LLMs lack ability to follow instructions for the diverse modules in MediQ–confidence estimation, abstention, asking questions–due to the fact that they are fine-tuned on instructions for QA tasks only [3].
>
> We will report Meditron (fine-tuned on Llama-2 with continued pre-training on medical texts then supervised fine-tuning on QA tasks) and Alpacare (Llama-2 based medical model instruction-tuned on synthetic medical data) on the BASIC and BEST setups and update the results in the paper, though based on our preliminary experiments, we expect poorer performance because these models are not as good at following general instructions. Overall, our main contribution is introducing the framework to study question-asking, using different base models for the Expert system will not impact the framework.
>
> We are grateful to the reviewer for their valuable feedback and believe these revisions will strengthen our paper. Thank you again for your valuable insights. We kindly request that the reviewer consider these points when assigning the final scores. We are happy to answer any further questions.
>
> ---
>
> [1] Li, Y., … & Zhang, Y. (2023). Chatdoctor: A medical chat model fine-tuned on a large language model meta-ai (llama) using medical domain knowledge. Cureus, 15(6).
>
> [2] Li, D., … & de Rijke, M. (2021, July). Semi-supervised variational reasoning for medical dialogue generation. 44th SIGIR (pp. 544-554).
>
> [3] Xie, Q., ... & Bian, J. (2024). Me llama: Foundation large language models for medical applications. arXiv:2402.12749.
>
> [4] Nori, H., … & Horvitz, E. (2023). Capabilities of gpt-4 on medical challenge problems. arXiv:2303.13375.
>
> [5] Chen, Z., ... & Bosselut, A. (2023). Meditron-70b: Scaling medical pretraining for large language models. arXiv:2311.16079.
>
> [6] Han, T., ... & Bressem, K. K. (2023). MedAlpaca--an open-source collection of medical conversational AI models and training data. arXiv:2304.08247.
>
> [7] Zhang, X., … & Petzold, L. R. (2023). Alpacare: Instruction-tuned large language models for medical application. arXiv:2310.14558.

---

### Official Review · Reviewer_M6dH · 2024-07-07

**Soundness:** 3
**Presentation:** 3
**Contribution:** 3
**Rating:** 7
**Confidence:** 4

**Summary:**

This paper introduce MediQ, a framework to simulate realistic clinical interactions, which incorporates a Patient System and an adaptive Expert System. The Patient system that simulates a patient and responds to follow-up questions, and an Expert system that serves as a doctor's assistant and asks questions to the patient before making a medical decision.

**Strengths:**

1. This paper proposes MediQ that stimulates realistic clinical interactions, which is more like a realistic scenario, rather than providing general responses at once.
2. The authors conduct extensive experiments to demonstrate the effectiveness of each component's design.

**Weaknesses:**

1. While the paper demonstrates its effectiveness via automatic evaluation metrics, it lacks human evaluation of the model's performance. Therefore, the performance of the system in "realistic" scenarios cannot be verified.
2. The response of the Expert system relies more on the parametric knowledge of LLMs. Although LLMs have learned knowledge during pre-training, their accuracy and stability cannot be guaranteed.

**Questions:**

See "Weaknesses".

**Limitations:**

See "Weaknesses".

---

> ### Author Rebuttal · Authors · 2024-08-07
>
> We thank the reviewer for their thorough review and feedback. We agree with the reviewer that our paper proposes a more realistic scenario for clinical interactions, and we conduct extensive experiments to validate each component of the framework. We hope to address the reviewer’s concerns below:
>
> > While the paper demonstrates its effectiveness via automatic evaluation metrics, it lacks human evaluation of the model's performance
>
> Thank you for highlighting this point. We will emphasize in our limitations section that extending MediQ to real human patients is not trivial due to factors such as data sensitivity, privacy risks, and linguistic and psychological adaptations to build trust with patients. As noted in the Ethics section, MediQ serves as an initial framework to enable development and evaluation of Expert systems rather than direct interaction with users (lines 415-416). We will clarify that the proposed interactive framework is “relatively more realistic” compared to static single-turn QA benchmarks, rather than deployable in real-life scenarios.
>
> We agree on the importance of human evaluations to further assess the model’s performance in such complex tasks. We conducted case-level interviews with an expert in clinical and medical education to get a sanity check on the model’s reasoning outputs. The expert noted that the model asks reasonable questions, but pointed out that questions in realistic clinical interactions can often be even more open-ended to elicit more detailed patient responses. We will include these insights and comments in the updated manuscript. While large-scale human annotations and testing on *real* patients to evaluate the framework’s real-life efficacy would require randomized control trials and present privacy and ethics concerns. However, conducting such detailed studies is beyond the scope of this paper due to the significant time and resources required. We will address this as a point for future work.
>
> **2. Parametric Knowledge:**
>
> > The response of the Expert system relies more on the parametric knowledge of LLMs. Although LLMs have learned knowledge during pre-training, their accuracy and stability cannot be guaranteed.
>
> We agree with the reviewer that the accuracy and stability of LLMs rely on parametric knowledge learned during pre-training, which is a shared concern among most works on LLMs. We will add this to the limitations section. However, we took several measures in our paper to reduce the effect of parametric knowledge:
>
> (1) Our abstention and rationale generation modules are explicitly designed to eliminate some of these effects by breaking down the complex clinical reasoning process into modularized components and providing more explainability. Rationale generation aims to provide more explainability into the knowledge gaps of the models compared to the standard QA setting where the model is only asked to select an option.
>
> (2) We also attempt to disentangle potential confounding variables by examining different combinations of model sizes (varying stability and knowledge), model families (potentially different pre-training data distribution), and abstention strategies. We found similar trends in the effects of different abstention strategies among different sizes and families, and observed statistical significant differences among the abstention strategies using the same model, thereby isolating the effects of confidence estimation from parametric knowledge.
>
> We also posit that models that can better rely on elicited knowledge rather than parametric would get better results in our interactive setting. Nonetheless, it's still a concern for real-deployment. Future work can explore how to augment relevant medical knowledge to ground the expert's questions and we will include the potential influence of pre-training in the limitations section.
>
> We are grateful to the reviewer for their valuable feedback and believe these revisions will strengthen our paper. Thank you again for your valuable insights. We kindly request that the reviewer consider these points when assigning the final scores. We are happy to answer any further questions.

---

> ### Comment · Reviewer_M6dH · 2024-08-13
> **Response to Authors**
>
> Thank you for your detailed response. I have raised my score accordingly.

---

### Official Review · Reviewer_SV4E · 2024-07-10

**Soundness:** 2
**Presentation:** 3
**Contribution:** 1
**Rating:** 5
**Confidence:** 3

**Summary:**

The paper proposes MediQ, a two-LLM-agent-system for interactive patient diagnosis. The authors argue that vanilla LLMs do not autonomously engage in an information extending discussion, but rather directly try to diagnose the patient and therefore oftentimes hallucinate.
The proposed system specifically aims to actively refrain from diagnosis when not sufficient information are available in order to ask more questions. This is embodied by a so-called expert agent. The patient agent, in turn, is prompted to support the expert agent by answering with relevant information.

The authors evaluate different versions of their system, where the patient agent either responds with the queried information or tries to further select relevant information, and the expert agent uses different abstention modules, e.g., predicting binary abstention, numeric confidences or including reasoning chains.
The evaluation is conducted for MEDQA and CRAFT-MD datasets, where patient information is partially removed to evaluate the interactive setting. The used LLM is GTP-3.5, showing that selecting relevant subset of patient information from the patient agent is superior to the easier strategies.
The authors also compare their approach to the full information setting for a number of additional base LLMs, including GPT-4 or Llama-3-8b. The results show that the interaction improves accuracy when compared to having only the multiple choice questions or base information about the compaint. The system, however, does not surpass the full information setting.
Finally, the author also provide several meta analyses, providing insights that, e.g., personalized abstention increases accuracy.

**Strengths:**

- Relevant topic: The paper deals with a relevant problem setting, trying to improve patient diagnosis using LLMs, where hallucination has bad consequences
- Sensible solution: The proposed system, using two agents which interact and support each other based on abstention and selecting relevant information, is well argued for.
- Good evaluation setup wrt data and base LLMs: The used datasets as well as the performed synthetic removal of patient information allow for a good evaluation of the problem. In addition, different base LLMs have been tested.
- Wide meta analysis: The conducted meta-analyses provide good insights, showing details about conversation lenghts and impacts of different strategies.

**Weaknesses:**

- Goal of the paper not clear: While the topic is relevant and the proposed abstention module is sensible, the paper sometimes refers to the outcome as simulation framework. Is the purpose then to evaluate other approaches through the simulation system rather than presenting a new abstention module?
- Related work not conclusive: It is not clear to me why mentioned works of the interactive models and agents section do not solve the same or an overlapping problem, which would make it necessary to evaluate against them. Mentioned works include [2, 18, 24], where one would have to argue better why the used approaches for improving interaction and sufficiently different or inapplicable here. In addition, mentioned competing medical diagnosis systen [45] is also not sufficiently described / exluded from being relevant for the evaluation, as only the abstention module seems to be missing. But it might be that the overall performance of used prompts / techniques even supersedes the reported ones here, right? Lastly, I do not understand the referencing of other papers promoting abstention during explaining the proposed method, i.e., [13], but not discussing the differences wrt novelty in the related work section.
- In addition to the covered works, I see an overlap to works such as [1], dealing with selective predictions of LLMs. It seems these methods have the same goal and should be compared to or discussed. A further elaboration is needed or a better
- As a consequence, at this point, I am not convinced the paper has sufficient novelty.
- Table 2 reports the reached accuracies of the different LLMs and variants. It seems from the table, that the full non-interactive setting is always paramount to the best interactive variant. It is not clear to me why abstention and patient information filtering then would help.

[1] Chen, Jiefeng, Jinsung Yoon, Sayna Ebrahimi, Sercan O. Arik, Tomas Pfister, and Somesh Jha. "Adaptation with self-evaluation to improve selective prediction in llms." arXiv preprint arXiv:2310.11689 (2023).
[2] Chinmaya Andukuri, Jan-Philipp Fränken, Tobias Gerstenberg, and Noah D. Goodman. Stargate: Teaching language models to ask clarifying questions, 2024.
[13] Shangbin Feng, Weijia Shi, Yike Wang, Wenxuan Ding, Vidhisha Balachandran, and Yulia Tsvetkov. Don’t hallucinate, abstain: Identifying llm knowledge gaps via multi-llm collaboration, 2024.
[18] Zhiyuan Hu, Chumin Liu, Xidong Feng, Yilun Zhao, See-Kiong Ng, Anh Tuan Luu, Junxian He, Pang Wei Koh, and Bryan Hooi. Uncertainty of thoughts: Uncertainty-aware planning enhances information seeking in large language models. arXiv preprint arXiv:2402.03271, 2024.
[24] Belinda Z Li, Alex Tamkin, Noah Goodman, and Jacob Andreas. Eliciting human preferences with language models. arXiv preprint arXiv:2310.11589, 2023.
 [45] Tao Tu, Anil Palepu, Mike Schaekermann, Khaled Saab, Jan Freyberg, Ryutaro Tanno, Amy Wang, Brenna Li, Mohamed Amin, Nenad Tomasev, Shekoofeh Azizi, Karan Singhal, Yong Cheng, Le Hou, Albert Webson, Kavita Kulkarni, S Sara Mahdavi, Christopher Semturs, Juraj Gottweis, Joelle Barral, Katherine Chou, Greg S Corrado, Yossi Matias, Alan Karthikesalingam, and Vivek Natarajan. Towards conversational diagnostic ai, 2024.


****** Update after author response *******
I thank the authors for their detailed clarifications. After reading the response and the other reviews, I am open to improve my score towards acceptance. I still think that it would be valuable and maybe a requirement that the framework includes better absention "modules", as there are papers who focus on that. The framework, in general, is an interesting and sensible interactive process, but to really add value the evaluation should better reflect that it is possible to gather all information or to abstain otherwise.

**Questions:**

-  Is the purpose of the paper to evaluate other approaches using the simulation framework rather than presenting a new abstention module?
- Can you please provide clearer discussions and argumentations for the mentioned related works wrt why the now proposed approaches are sufficiently different / have a different goal / do not need to be evaluated against them?
- How does the proposed abstention approach relate to [1]?
- Why does the full non-interactive setting reach the best overall accuracy even though hallucination takes place?

**Limitations:**

Limitations have been discussed in the paper, but I wonder what would happen if the patient LLM would make up facts and how this might be confidently tested at real-time. If relevant, this could be added to the discussion.

---

> ### Author Rebuttal · Authors · 2024-08-07
>
> We appreciate the reviewer’s feedback and for highlighting our strengths, including the relevance of the topic, the efficacy of the proposed interactive system, and our thorough evaluation and analysis.
>
> > Is the purpose then to evaluate other approaches through the simulation system rather than presenting a new abstention module?
>
> Thank you for the question. The main contribution of our paper is both the simulation/evaluation framework and insights from comparing different abstention modules, rather than proposing a state-of-the-art abstention method.
>
> Our goal is to study and improve LLMs in medical diagnostics, where eliciting information via interactions with patients are crucial. We introduce a two-agent interactive setup to evaluate the “information seeking” ability of LLMs (ln.5-6,34-40,43,106,393-396) and then focus on improving this behavior by enabling the model to know when to ask questions.
>
> - Our evaluation framework is a paradigm shift from the standard single-turn QA setup where all information is given upfront. Instead, the Expert only starts with some initial information, and elicits more info as it continues to interact. We posit that this is more realistic and challenging and is essential in building medical AI assistants. We will better motivate the interactive setup in the paper.
>
> - Second, we focus on abstention, investigating how knowing when to ask questions impacts information-seeking (ln.54,149-152). We combine established confidence estimation techniques in novel ways (ln.155-177). While the techniques are not exhaustive, we found that better confidence estimation leads to better diagnostic performance. We close the gap between the more realistic information-seeking paradigm to the full information setup by 45.5% (ln.296-297), providing a starting point for research in proactive information seeking. Our fine-grained analysis offers insights regarding the **conversation length** (§4.3), **relevance of the questions** (ln.307-316), **format of the information** (ln.317-321), **confidence thresholding** (ln.337-348), and **rationale generation** (ln.349-365) to guide future research.
>
> > related works wrt why the proposed approaches are sufficiently different / have a different goal / do not need to be evaluated against them
>
> We acknowledge the importance of [2, 18, 24, 45, 13] and clarify that our goal is not to propose a new confidence estimation approach but to show that different abstention methods can work with our interactive framework. We now explain our comparisons and why some were not included.
>
> - [24] elicits user preferences on common tasks but lacks abstention. Our interactive baseline--Basic (ln.155-156)--adapts [24] into MediQ, which we show is not sufficient for clinical reasoning. In MediQ, we adapt some ideas and prompts from [24] but in addition incorporated abstention, as well as complex clinical reasoning tasks and domain knowledge prompts, making our framework more relevant for medical reasoning.
>
> - [2] focuses on how to ask good questions via RL-based training; [18] studies how to select questions. They both differ from our focus on *when* to ask questions via abstention at inference time. We don’t compare to [2] and [18] because they study a different problem than ours and require training [2] or yes/no questions [18].
>
> - [45] lacks an abstention module as noted by the reviewer. [45] involves fine-tuning a *closed-source model on a closed-source dataset*, making it impossible to reproduce. Future work could integrate [45]’s modules into MediQ to enhance the Expert system, but we can’t compare it in our paper due to the need for training and lack of access.
>
> - [13] presents confidence estimation methods for abstention in general tasks *without* interaction. We adapted some of these prompt-based methods as baselines (Numerical Abstention in MediQ corresponds to “Ask for Calibration” in [13]).
>
> - [1] is a confidence estimation method focusing on selective prediction, different from our work on *question-asking* and *information-seeking* in interactive medical dialogues. While we can see [1] being adopted as an Abstention strategy, we don’t compare it in MediQ due to the need for training and model probabilities to calculate confidence scores.
>
> Our work, while acknowledging and building on [2, 18, 24, 45, 13], introduces novel contributions focused on **when to ask questions** via abstention at inference time to ensure safe and effective medical dialogue systems. Our novelty also lies in adapting existing confidence estimation methods into the medical domain, which requires more complex reasoning pathways and domain knowledge and was not explored by the above works. We will clarify these points in the paper and include [1] in the related work section.
>
> > Why does the full non-interactive setting reach best overall accuracy even though hallucination takes place?
>
> The full non-interactive setup gives all information upfront, so the Expert only has to process the already sufficient information and produce a diagnosis (ln.20). This setup is unrealistic in practice (ln.19,30-33), so we propose the paradigm shift to only provide partial information initially (ln.9,76-77). The Expert is now tasked with 1) deciding when to ask questions, 2) elicit patient information, and 3) integrate information to produce diagnosis, which is inherently a harder task. Starting from only the chief complaint, the interactive Expert aims to elicit patient information to match the full non-interactive upper bound (ln.20,57-58). The reviewer is correct that the best interactive Expert still lags behind the full non-interactive upper bound, but this gap highlights the need for further development in proactive information-seeking LLMs (ln.19-20,296-298).
>
> Thank you again for the detailed suggestions and hope we cleared up potential confusions. We kindly request that these points be considered when assigning the final scores. We are happy to answer any further questions.

---

> > ### Comment · Reviewer_SV4E · 2024-08-12
> > **Thanks for the clarifications**
> >
> > I thank the authors for their answers to my questions, and understand parts of the motivation and value better now. I have no further questions at this point.

---

> > > ### Author Response · Authors · 2024-08-12
> > >
> > > We welcome further questions about our work, and would really appreciate an appropriate increase in score if the reviewer’s concerns are adequately addressed. Thank you!

---

### Official Review · Reviewer_5NqX · 2024-07-14

**Soundness:** 3
**Presentation:** 3
**Contribution:** 3
**Rating:** 7
**Confidence:** 3

**Summary:**

The goal of the paper is to develop a dataset on which models can be trained for interactive LLM-patient dialogues that require follow-up questions. The paper adapts MedQA and CraftMD datasets into an interactive setup, uses an LLM to mimic a patient's questions and trains LLMs to ask the necessary follow-ups to answer the patient's primary question.

**Strengths:**

Strengths (originality, quality, clarity, and significance)

Adapting existing datasets into a conversational setup is a useful contribution
that will likely be used in future work.

The paper is clearly written.

The paper evaluates on both closed-source and open-source LLMs.

The finding that a naive question-asking approach actually decreases
performance is quite interesting.

The experimental evaluation of different methodologies for improved follow-up
questions and abstention is thorough and convincing.

**Weaknesses:**

The interactive framework means that the evaluation of Expert systems is
heavily dependent on the idiosyncracies of the Patient system.  While it is
clear that different choices of Patient and Expert systems have meaningful
impacts on the overall QA performance, the evaluation is not detailed enough to
understand whether performance on MedIQ would generalize to performance for
human patients. While this limitation is perhaps obvious, it should be
addressed more explicitly and it would be helpful to guide future work to
discuss how the authors would ideally expand this work to include human
patients and/or clinicians. In particular, if the Expert systems are able to
answer the QA tasks with information that would not be sufficient for a human
clinician, that may be a bug rather than a feature.

**Questions:**

Suggestion: I appreciate the thoroughness of the experimental analyses but think you are
trying to fit too much content into too few pages. The font size and spacing of
figures are inaccessibly small, and Figures (e.g., 6) are too crowded to easily
extract the important takeaways. I think it would be better to try to
streamline the main body of the paper and add a few more pages to the appendices.

**Limitations:**

See weakness above.

---

> ### Author Rebuttal · Authors · 2024-08-07
>
> We thank the reviewer for the insightful feedback and thorough review for our paper. We appreciate the reviewer highlighting the strengths of our paper, including adapting existing datasets into conversation format, clarity, interesting findings and thorough experiments and models. We address the reviewer’s concerns and questions below:
>
> > the evaluation is not detailed enough to understand whether performance on MedIQ would generalize to performance for human patients.
>
> Thank you for raising this concern. Generalization to human patients is indeed important and non-trivial, so we leave it for future work. We will include a discussion in the updated manuscript on the challenges with extending MediQ to human patients as well as important areas for future extensions of this framework. We briefly outline a few ideas here:
>
> - a) Incorporating complex **human and social factors** into the Patient system to simulate real-world interactions accurately. For instance, patients might omit information regarding stigmatized diseases, and symptoms might present differently in different patients.
>
> - b) Exploring how to interact with human patients using MediQ from **linguistic and pragmatics perspectives**. This includes, studying conversational nuances, trust building, communication style, and culturally specific norms to bring the framework closer to real-world application.
>
> - c) Train a patient system on actual conversation data from real patients to better simulate interactions with a real human patient. Collection of such clinical interaction datasets, although tricky and potentially containing privacy risks, would greatly benefit the community.
>
> - c) Conducting **randomized control trials** with patients to test the real-life efficacy of the framework. However, conducting such detailed and careful studies is not feasible within the scope of this paper, as it requires significant time, resources, and medical expertise.
>
> Additionally, we agree with the reviewer that the Patient system is critical for evaluating the Expert system. Our preliminary experiments with different variants of a Patient showed that irrelevant responses or factually inaccurate responses from the patient led to lower diagnostic accuracy. Therefore, we focused on evaluating the factuality and relevance of a Patient system's responses, to ensure high quality, relevant responses to an Expert's question (Section 3.1).
>
> > it would be better to try to streamline the main body of the paper and add a few more pages to the appendices
>
> Thank you for the feedback on space and organization of the paper. We will incorporate your suggestions and shorten sections 2 and 3 (framework description and methods), move the analysis of the abstention threshold (line 337-348, Figure 8(a)) to the appendix, and make sure all figures are legible and highlight important takeaways.
>
> We appreciate your constructive feedback and believe these revisions will strengthen our paper. Thank you again for your valuable insights.

---

> > ### Comment · Reviewer_5NqX · 2024-08-11
> >
> > I appreciate the response. I think the discussion of extensions to human patients would be a welcome addition.

---

### Decision · Program_Chairs · 2024-09-25

**Decision:**

Accept (poster)

**Comment:**

This paper proposes an interactive approach to medical question answering in which the LLM module is designed to ask follow-up questions that elicit missing information necessary to inform an accurate response. This system entails design of an abstention module which determines whether to seek additional information from a patient via follow-up questions. The results suggest that allowing LLMs to interactively seek additional information when helpful appears to boost model performance.

Perhaps the main limitation of this work is the lack of manual evaluation, which the authors acknowledged in the response period. But despite this, the work offers a novel information-seeking framework for integrating LLMs into medical prediction models, and may influence such efforts going forward.